# Understanding VPAC receptor family peptide binding and selectivity

Sarah J. Piper [1,2], Giuseppe Deganutti [3], Jessica Lu[1,2], Peishen Zhao [1,2], Yi-Lynn Liang[1,7], Yao Lu[1,2], Madeleine M. Fletcher [1,8], Mohammed Akhter Hossain [4], Arthur Christopoulos [1,2], Christopher A. Reynolds [3,5], Radostin Danev [6], Patrick M. Sexton [1,2] ✉ & Denise Wootten [1,2] ✉

The vasoactive intestinal peptide (VIP) and pituitary adenylate cyclase-activating polypeptide (PACAP) receptors are key regulators of neurological processes. Despite recent structural data, a comprehensive understanding of peptide binding and selectivity among different subfamily receptors is lacking. Here, we determine structures of active, Gs-coupled, VIP-VPAC1R, PACAP27-VPAC1R, and PACAP27-PAC1R complexes. Cryo-EM structural analyses and molecular dynamics simulations (MDSs) reveal fewer stable interactions between VPAC1R and VIP than for PACAP27, more extensive dynamics of VIP interaction with extracellular loop 3, and receptor-dependent differences in interactions of conserved N-terminal peptide residues with the receptor core. MD of VIP modelled into PAC1R predicts more transient VIP-PAC1R interactions in the receptor core, compared to VIP-VPAC1R, which may underlie the selectivity of VIP for VPAC1R over PAC1R. Collectively, our work improves molecular understanding of peptide engagement with the PAC1R and VPAC1R that may benefit the development of novel selective agonists.

The pituitary adenylate cyclase-activating polypeptide (PACAP) receptor subfamily consists of three class B1 G protein-coupled receptors (GPCRs) that have diverse physiological functions, including modulation of inflammatory and neuroprotective signalling pathways[1–3]. The PACAP type 1 receptor (PAC1R) is activated by PACAP and vasoactive intestinal peptide (VIP), however, PACAP has approximately 1000-fold higher affinity than VIP[4]. In contrast, the VIP/PACAP receptors 1 (VPAC1R) and 2 (VPAC2R) have high affinity for both PACAP and VIP and are activated with high potency by both peptides[5]. PACAP exists in two isoforms, PACAP38, a 38 amino acid peptide, and PACAP27, a C-terminal truncated 27 amino acid isoform[6,7]. These peptides have high homology with VIP, sharing nearly 70% sequence identity (Fig. 1a), and the PAC1R and VPAC1R also exhibit a high degree of sequence homology (56% overall (https://gpcrdb.org)), which presents significant challenges for developing selective drugs. Thus, there is significant interest in understanding the molecular basis for peptide selectivity between the PAC1R and VPACRs.

A hallmark of class B1 GPCRs is a relatively large N-terminal extracellular domain (ECD) of ~150 amino acids that is the site of initial binding of the C-termini of peptide agonists, and this facilitates subsequent engagement of the peptide N-terminal activation domain with the extracellular loops (ECLs) and transmembrane helices (TMs) of the receptor core[8,9]. Structural, biochemical and computational studies have revealed that the ECD of class B1 GPCRs is highly mobile, even

[1]Drug Discovery Biology, Monash Institute of Pharmaceutical Sciences, Monash University, Parkville 3052 VIC, Australia. [2]ARC Centre for Cryo-electron Microscopy of Membrane Proteins, Monash Institute of Pharmaceutical Sciences, Monash University, Parkville 3052 VIC, Australia. [3]Centre for Sport, Exercise and Life Sciences, Coventry University, CV1 5FB Coventry, UK. [4]Florey Institute of Neuroscience and Mental Health, The University of Melbourne, Parkville, VIC 3010, Australia. [5]School of Life Sciences, University of Essex, Wivenhoe Park, Colchester CO4 3SQ, UK. [6]Graduate School of Medicine, University of Tokyo, S402, 7-3-1 Hongo, Bunkyo-ku, 113-0033 Tokyo, Japan. [7]Present address: Confo TherapeuticsTechnologiepark 94, Ghent (Zwijnaarde) 9052, Belgium. [8]Present address: GlaxoSmithKline, Abbotsford 3067 VIC, Australia. ✉e-mail: patrick.sexton@monash.edu; denise.wootten@monash.edu

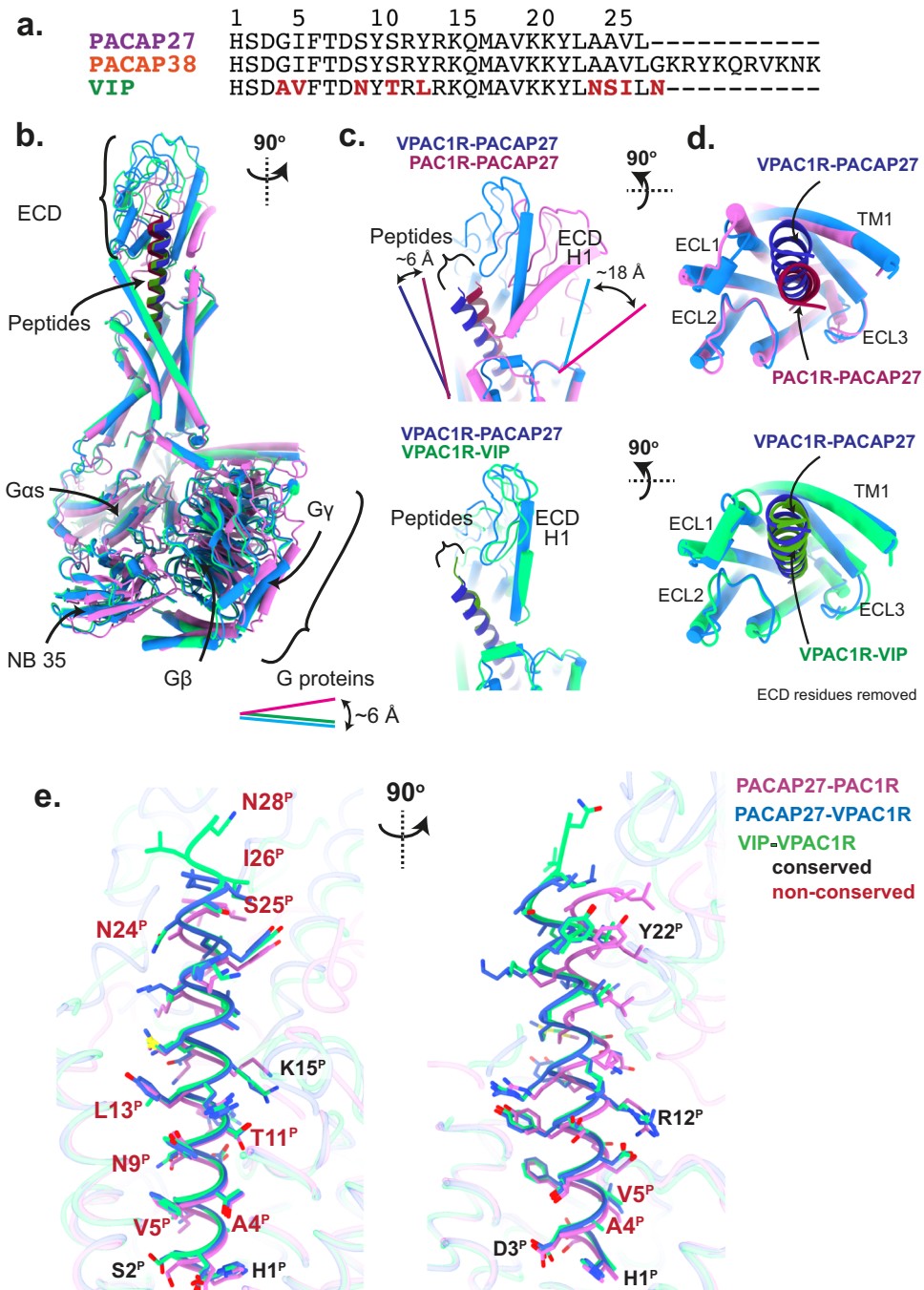

**Fig. 1 | Overview of active, Gs-coupled PACAP family complexes solved using cryo-EM. a** Sequence alignment of endogenous PACAP family peptides PACAP27, PACAP38 and VIP, with non-conserved VIP residues shown in red. **b**–**d** Secondary structure comparison of Gs complexes of PAC1R-PACAP27 (pink, peptide: dark red), VPAC1R-VIP (light green, peptide: dark green) and VPAC1R-PACAP27 (light blue, peptide: dark blue). Secondary structure is shown as ribbon for the peptide helix and as cylinders (helices) and beta sheets for the rest of the complex. Complexes are shown in different angles as front view of the entire complex (**b**, all three structures), side view of the extracellular domain (ECD) and peptide C-terminus (**c**, PACAP27 bound structures top panel, VPAC1R structures bottom panel) and top view of the receptor bundle, extracellular loops (ECL) and peptide C-terminus

(**d**, PACAP27 bound structures top panel, VPAC1R structures bottom panel, with ECD residues removed for clarity). Offsets between VPAC1R and PAC1R structures of ECD helix 1 (H1, C-terminus), G proteins (Gs αN N-terminus) and peptides (C-terminus) are indicated by lines with distances reported in Å (distance measured between Cα atoms of terminal residues). **e** Overview of peptide residues within the receptor complexes (receptor-aligned), shown as front and side view (left and right, respectively) with receptor backbone displayed in transparent ribbon format and peptide backbone in ribbon format with sidechains displayed in stick format. Residues labelled in red are VIP residues not conserved with PACAP (as noted in (**a**)).

when bound to peptides that span the ECD and TM core. Consequently, the ECD is poorly resolved in most published active state class B1 GPCR cryogenic electron microscopy (cryo-EM) structures[9,10], limiting understanding of key facets of peptide binding and the activation processes of this receptor subfamily.

Cryo-EM derived, active, Gs protein-coupled, structures of the PACAP receptor subfamily have been determined for VPAC1R bound to PACAP27[10] and for PAC1R coupled to PACAP38[9] or maxadilan[11]; the latter an exogenous PAC1R-selective agonist that shares no structural similarities with PACAP or VIP[12]. The previously published cryo-EM

**Table 1 | Map and model information for VPAC1R-PACAP27, VPAC1R-VIP and PAC1R-PACAP27 datasets**

| | VPAC1R-VIP-Gs | VPAC1R-PACAP27-Gs | PAC1R-PACAP27-Gs |
|---|---|---|---|
| **Data collection/processing** | | | |
| Magnification | 105 000x | 130 000x | 105 000x |
| Voltage (kV) | 300 | 300 | 300 |
| Spot size | 4 | 7 | 5 |
| Total electron dose (e⁻/Å²) | 64.9 (set1) 47 (set2)* | 69.7 | 52.4 |
| Exposure time (s) | 3.196 | 6.00 | 5.01 |
| Energy filter slit width (eV) | 25 | 15 | 25 |
| Movie frames | 80 | 85 | 71 |
| Dose rate (e-/Å²/s) | 20.3 (set1) 14.7 (set2)* | 11.62 | 10.45 |
| Movies collected | 1134 (set1) 4329 (set2)* | 6745 | 5850 |
| K3 CDS mode | No | Yes | Yes |
| Defocus range (µm) | −0.5 to −1.4 | −0.5 to −1.4 | −0.5 to −1.4 |
| Pixel size (Å) | 0.83 | 0.65 | 0.83 |
| Final particle number | 375 164 | 334 267 | 440 740 |
| **Relion maps (post-processed)** | | | |
| Resolution (Å) | 2.7 (a), 2.9 (b), 3.0 (c) | 2.3 (a), 2.4 (b), 2.5 (c) | 2.3 (a), 2.4 (b), 2.5 (c) |
| Map sharpening B-factor (Å²) | −60 (auto), −20 (man) | −69 (auto), −25 (man) | −96 (auto), −25 (man) |
| Map vs Model (Phenix Validation) | 0.77 | 0.85 | 0.82 |
| **Cryosparc maps** | | | |
| Resolution (Å) | 2.8 | 2.4 | 2.4 |
| **Validation molprobity** | | | |
| MolProbity score (percentile) | 1.45 (96th) | 1.31 (98th) | 1.35 (98th) |
| Clashscore (percentile) | 7.18 (86th) | 5.72 (91th) | 5.78 (91th) |
| Poor rotamers (%) | 0.11% | 0.11% | 0% |
| **Ramachandran plot** | | | |
| Favoured (%) | 97.75 | 98.04 | 97.86 |
| Outliers (%) | 0 | 0 | 0 |
| Cβ deviations (%) | 0 | 0 | 0 |
| **Bonds** | | | |
| Bond length (Å), bad bonds (%) | 0.003, 0% | 0.003, 0% | 0.003, 0% |
| Bond angles (°), bad angles (%) | 0.588, 0.02% | 0.599, 0.02% | 0.582, 0.06% |

Changed imaging conditions during data collection of VPAC1R-VIP dataset (labelled as set1/set2, indicated by asterisk*. Initially processes separately, then re-joined refined particles. (a) Resolution (Å) of consensus map excluding micelle/AHD/ECD (tight mask). (b) Resolution (Å) of consensus map including micelle (wide mask). (c) Resolution (Å) of receptor-focused cryo-EM map. (auto) Auto-B-factor of high-resolution map (water molecules visible) (deposited as additional map). (man) Manual B-factor of consensus map.

structures have not been sufficient to rationalize the selectivity of VIP for the VPACRs over the PAC1R. The poor resolution of the ECDs in the published data suggests that this domain is highly mobile and thus the dynamics of the agonist-bound complexes may play a key role in the binding and selectivity of peptides, as has been seen for other class B1 GPCRs[13–15]. Here we present cryo-EM structures and conformational variance analysis of VIP-VPAC1R-Gs, PACAP27-VPAC1R-Gs and PACAP27-PAC1R-Gs complexes. In conjunction with molecular dynamics simulations (MDSs), we reveal critical differences in the interactions and dynamics of peptides with VPAC1R and PAC1R and

provide a framework for understanding peptide selectivity at these receptors.

## Results

The PAC1R exists in different splice isoforms with a deletion within the ECD termed PAC1Rshort, and insertions within ICL3, termed PAC1Rhip, PAC1Rhop, PAC1Rhiphop[16]. PAC1Rnull does not have ICL3 insertions and contains additional residues in the ECD loop (residues 89–110) compared to PAC1Rshort[16]. For this study, PAC1R refers to the PAC1Rnull variant. This was selected as this is the most abundantly expressed variant physiologically, is the most well-studied, and the lack of ICL3 insertions provides the best comparison both structurally and pharmacologically to VPAC1R. This variant also has greater peptide selectivity of PACAP27 and PACAP38 over VIP relative to PAC1Rshort[16], and is therefore an ideal choice to provide insights into the selectivity of PAC1R for PACAP27 over VIP.

### Cryo-EM structures of Gs-coupled PAC1R and VPAC1R bound to endogenous ligands

Following expression, complex formation and purification (Supplementary Fig. 1), cryo-EM maps were reconstructed for each receptor complex (Supplementary Fig. 2, Table 1). The VIP-VPAC1R-Gs complex cryo-EM maps were refined to global resolutions of 2.7 Å and 2.9 Å (tight and wide mask, respectively), while focused refinement of the receptor reached 3.0 Å, according to the gold-standard FSC at the 0.143 criterion (Supplementary Fig. 2g–j). The datasets containing complexes with PACAP27 (VPAC1R-Gs and PAC1R-Gs), were each refined to global resolutions of 2.3 Å or 2.4 Å (tight and wide mask, respectively), and the receptor alone refinement reached 2.5 Å (Supplementary Fig. 2h, i, k, l, Table 1). The cryo-EM maps allowed for confident assignment of the backbone and side chain rotamers for the majority of each of the complexes (Supplementary Fig. 3a–c), with the exception of the ECD, where the backbone was modelled for all complexes (except for the PAC1n ECD loop and stalk region), and Gs protein α-helical domains (AHD) that were omitted from models due to lower resolution.

### Peptide binding to PAC1R and VPAC1R in consensus cryo-EM structures

The secondary structures of receptor TMs, peptides and G proteins are in overall high agreement with other published class B1 GPCR structures[14,17]. Both peptides (VIP and PACAP27) exhibited an extended helical structure regardless of the receptor with the peptide N-termini buried in the TM bundle and the peptide C-termini interacting with ECD residues (Fig. 1b–e). Comparing VPAC1R and PAC1R, there were receptor-dependent offsets in the locations of the peptide C-termini, the receptor ECDs, and the bound Gs proteins (Fig. 1b–d). VIP and PACAP27 bound to VPAC1R are angled towards TM1/TM2 as they exit the receptor core, whereas PACAP27 bound to PAC1R exits the core in a vertically aligned orientation (-6 Å/-8 degrees relative to PACAP27-VPAC1R). These distinctions in peptide orientation are aligned with the tilt position of the ECDs (Fig. 1c) where helix 1 (H1ᴱᶜᴰ) of the PAC1R ECD is tilted back (-18 Å/-35 degrees), relative to H1ᴱᶜᴰ of VPAC1R that is oriented closer to perpendicular to the membrane plane. PAC1R also displayed a shorter TM1 helix (-one turn), with poorly resolved map density for the stalk region that links TM1 and the ECD, compared to the VPAC1R structures.

The receptor TMs and the peptide N-termini are the highest resolution areas of the receptor-peptide complex, enabling modelling of structured waters and detailed investigation of contact networks. Overall, the peptide N-termini formed more polar bonds and hydrophobic contacts with the receptors than the peptide C-termini (Fig. 2, Supplementary Fig. 4). Multiple polar contacts were conserved for both receptors, including H-bonds between S11ᴾᴬᶜᴬᴾ or T11ⱽᴵᴾ with ECL2 residues D287ⱽᴾᴬᶜ¹ᴿ/D298ᴾᴬᶜ¹ᴿ, and T7ᴾᴬᶜᴬᴾ/ⱽᴵᴾ with K2.67ᴾᴬᶜ¹ᴿ/ⱽᴾᴬᶜ¹ᴿ

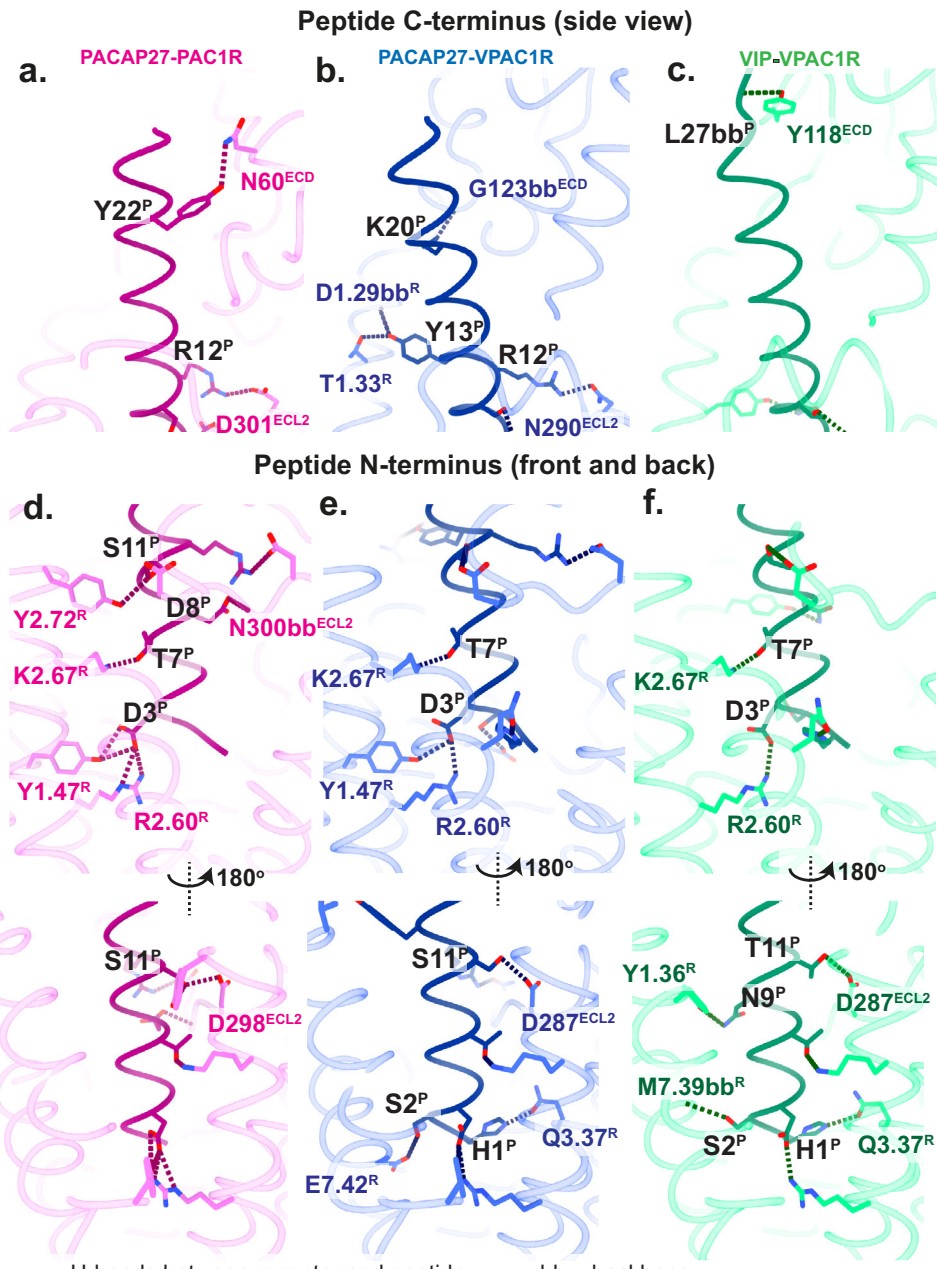

**Fig. 2 | Hydrogen bonds between peptide (P) and receptor (R) residues in the static structures.** The receptor backbone is displayed in transparent in ribbon format and peptide backbone in ribbon format, with sidechains that are involved in H-bonds displayed in stick format. PAC1R-PACAP27 is shown in pink-dark pink, VPAC1R-PACAP27 in blue-dark blue and VPAC1R-VIP in green-dark green. **a–c** Interactions of the peptide C-terminus and the receptor shown as side view.

**d–f** Interactions of the peptide N-terminus and the receptor shown as front and back view. H-bonds between peptide and receptor are displayed as dotted lines. H-bonds involving backbones, and not sidechains, are labelled as 'bb'. Receptor residues are numbered according to the Wootten et al.[18], class B1 scheme. Residues in the ECL and ECD as well as peptide residues are labelled according to the residue number.

(receptor residues are numbered according to the Wootten et al., class B1 receptor numbering scheme; the first digit refers to the helix location[18]). There was also a conserved salt bridge between D3[PACAP/VIP] and R2.60[PAC1R/VPAC1R] (Table 2, Supplementary Fig. 4). Interactions in the structures that differed between the peptides or across the receptors for PACAP27 occurred predominantly in the C-terminal half of the peptide (Fig. 2a–c; Supplementary Fig. 4). VIP engages in fewer H-bonds with VPAC1R (seven H-bonds) than PACAP27 (ten H-bonds for both receptors) (Table 2) and also has fewer contacts between the peptide mid-/C-terminal half and the receptor (as shown by the absence of, or fewer contacts of R12[VIP], L13[VIP], K20[VIP]) (Fig. 2c, Supplementary Fig. 4).

The cryo-EM map resolution, particularly for the PACAP27-bound receptors, enabled clear visualization of ordered water molecules in the peptide binding pocket and G protein interface. A few conserved ordered water molecules within the PAC1R and VPAC1R TM core are present in all structures, irrespective of the bound peptide (for example in the vicinity of S5.46 and E3.50). These are also conserved in the glucagon-like peptide 1 (GLP-1) receptor (GLP-1R) bound to GLP-1 (PDB: 6X18)[17], showing conservation of waters across class B1 receptors. Nonetheless, there were receptor-dependent water networks in the vicinity of peptide residues H1-S2-D3, residues critical for agonist potency[19–21]. The S2[PACAP27] backbone and side chain penetrate deeper into the PAC1R TM bundle compared to S2[PACAP27/VIP] in VPAC1R,

**Table 2 | Hydrogen bonds predicted by ChimeraX between receptor and peptide chains and receptor and G protein chains**

| Receptor - Peptide | | | | | | | | |
|---|---|---|---|---|---|---|---|---|
| **PAC1R-PACAP27** | | | **VPAC1-PACAP27** | | | **VPAC1-VIP** | | |
| **Peptide** | **Receptor** | **Distance (Å)** | **Peptide** | **Receptor** | **Distance (Å)** | **Peptide** | **Receptor** | **Distance (Å)** |
| D3 | Y1.47 | 3.34 | H1 | Q3.37 | 2.65 | H1 | Q3.37 | 3.16 |
| D3 | Y1.47 | 3.36 | S2 | E7.42 | 3.29 | S2 | M7.39 | 3.37 (bb) |
| D3 | R2.60 | 3.31 (s) | D3 | Y1.47 | 3.36 | D3 | R2.60 | 2.87 (s) |
| D3 | R2.60 | 2.55 (s) | D3 | R2.60 | 2.94 (s) | T7 | K2.67 | 2.46 |
| T7 | K2.67 | 2.50 | T7 | K2.67 | 2.51 | N9 | Y1.36 | 3.38 |
| D8 | N300$^{ECL2}$ | 2.86 | S11 | D287$^{ECL2}$ | 2.93 | T11 | D287$^{ECL2}$ | 3.01 |
| S11 | Y2.72 | 3.18 | R12 | N290$^{ECL2}$ | 2.99 | L27 | Y118$^{ECD}$ | 2.35 (bb) |
| S11 | D298$^{ECL2}$ | 2.91 | Y13 | D1.29 | 3.20 (bb) | | | |
| R12 | D301$^{ECL2}$ | 2.90 (s) | Y13 | T1.33 | 2.77 | | | |
| Y22 | N60$^{ECD}$ | 3.27 | K20 | G123$^{ECD}$ | 3.43 (bb) | | | |
| **Receptor – G proteins** | | | | | | | | |
| **PAC1R-PACAP27-Gs** | | | **VPAC1-PACAP27-Gs** | | | **VPAC1-VIP-Gs** | | |
| **Gs** | **Receptor** | **Distance (Å)** | **Gs** | **Receptor** | **Distance (Å)** | **Gs** | **Receptor** | **Distance (Å)** |
| D381 | K5.64 | 2.56 | N384 | L244$^{ICL2}$ | 2.83 | N384 | L244$^{ICL2}$ | 2.64 |
| N384 | L255$^{ICL2}$ | 2.75 | N384 | K5.64 | 2.93 | R385 | K5.64 | 3.54 (bb) |
| N384 | K5.64 | 2.87 | R385 | K5.64 | 3.27 (bb) | R380 | S247 | 3.41 |
| R385 | K5.64 | 3.10 (bb) | R385 | K5.64 | 3.48 (bb) | N384 | K5.64 | 2.78 |
| E392 | G405$^{H8}$ | 2.72 | Y391 | H1.50 | 3.20 | D323 | R329$^{ICL3}$ | 3.08 |
| L393 | S6.41 | 3.16 | E392 | G393$^{H8}$ | 2.91 | E392 | G393$^{H8}$ | 2.82 |
| | | | L393 | S6.41 | 2.73 | | | |
| **Gβ** | **Receptor** | **Distance (Å)** | **Gβ** | **Receptor** | **Distance (Å)** | **Gβ** | **Receptor** | **Distance (Å)** |
| H311 | R413$^{H8}$ | 2.43 | n/a | | | n/a | | |
| D312 | R413$^{H8}$ | 3.15 | | | | | | |

Hydrogen bonds were predicted using relaxed parameters and distances are shown in Å. (s) indicates a salt bridge. (bb) indicates interactions with the backbone. n/a: no hydrogen bonds present. Receptor residues are labelled according to the Wootten et al. numbering system.

overlapping a site occupied by water molecules in both peptide-bound VPAC1R structures. While the S2 side chain in both peptides engages with the conserved E7.42$^{VPAC1R/PAC1R}$ in both receptors, the additional space in the binding pocket in the vicinity of S2 observed in the VPAC1R structures led to a distinct water network around the N-terminal peptide residues relative to PAC1R structures (Supplementary Fig. 5).

Recent cryo-EM structures of PAC1R-PACAP38 (PDB: 6M1I[11], PDB: 6LPB[22], PDB: 6P9Y[9]) at 3.5 Å, 3.9 Å and 3.01 Å, respectively, are in overall agreement with the presented PAC1R-PACAP27 in terms of secondary structure comparison of the models built into the cryo-EM maps (Supplementary Table 1). However, previous cryo-EM maps lack the resolution to model water molecules, and extracellular domains and loops could also not be modelled (Supplementary Fig. 6a). The published structure of VPAC1R-PACAP27 at 3.2 Å (PDB 6VN7) used the Nanobit technology to stabilize the G protein complex[10], and the model for this also aligns well at the secondary structure level with our VPAC1R-PACAP27 structure (Supplementary Fig. 6b).

### Receptor-peptide dynamics: CryoSPARC 3DVA and equilibrium MD simulations

Previously published structures of the VPAC receptor family lack dynamic information that can be gained from cryo-EM datasets. Using the particle stacks from our cryo-EM consensus refinements, we applied 3D variability analysis (3DVA) in cryoSPARC to compare the overall dynamics of receptor-G protein complexes and assessed differences between the peptide-receptor pairs (Supplementary Movies 1, 2, Supplementary Fig. 7). All complexes exhibited rocking and twisting motions between receptor and the bound G protein, similar to those

previously described for 3DVA of other class-B1 GPCR-G protein cryo-EM datasets[13,14] (Supplementary Movies 1, 2). Of note, the rotational and translational offsets of the peptide and G protein, observed between different receptors in the static consensus structures, were partially sampled within the 3DVA frames (Supplementary Movie 1, Supplementary Fig. 7). For this analysis, we focused on the dynamics within and between the receptor and peptide chains by comparing models of the backbones from the consensus structures that were placed into the 3DVA maps corresponding to the extreme frames from the trajectories (frame 0 and 19) of three principal components (components 0 to 2) (Supplementary Movie 2, Fig. 3a, b, Supplementary Fig. 7). These revealed that for all experimental structures the peptide N-terminus bound within the TM bundle was relatively stable, whereas the peptide C-terminus was dynamic, moving in concert with the ECD (Supplementary Movie 2). While ECL3 in the VIP-VPAC1R complex adopted a slightly more open conformation in all 3DVA frames, compared to the PACAP bound PAC1R and VPAC1R complexes, ECL3 was also dynamic and is consistent with the consensus cryo-EM maps displaying low resolution of this loop in this complex (Fig. 3a).

To further probe the conformational dynamics, equilibrium MDSs were performed on each of the complex structures using the models built into the experimental consensus cryo-EM maps as the starting template (Fig. 3c–e, Supplementary Table 2), and these were compared to the cryo-EM 3DVA (Fig. 3a, b). For visualization, snapshots from the MDSs were extracted as frames (every 100th frame). The most striking conformational differences were apparent in the VIP-VPAC1R structure that had the greatest range of motion, particularly for the peptide C-terminus and for ECL3 and the adjacent TM1 (Fig. 3e). By comparison, PACAP27, bound to either PAC1R or VPAC1R, had

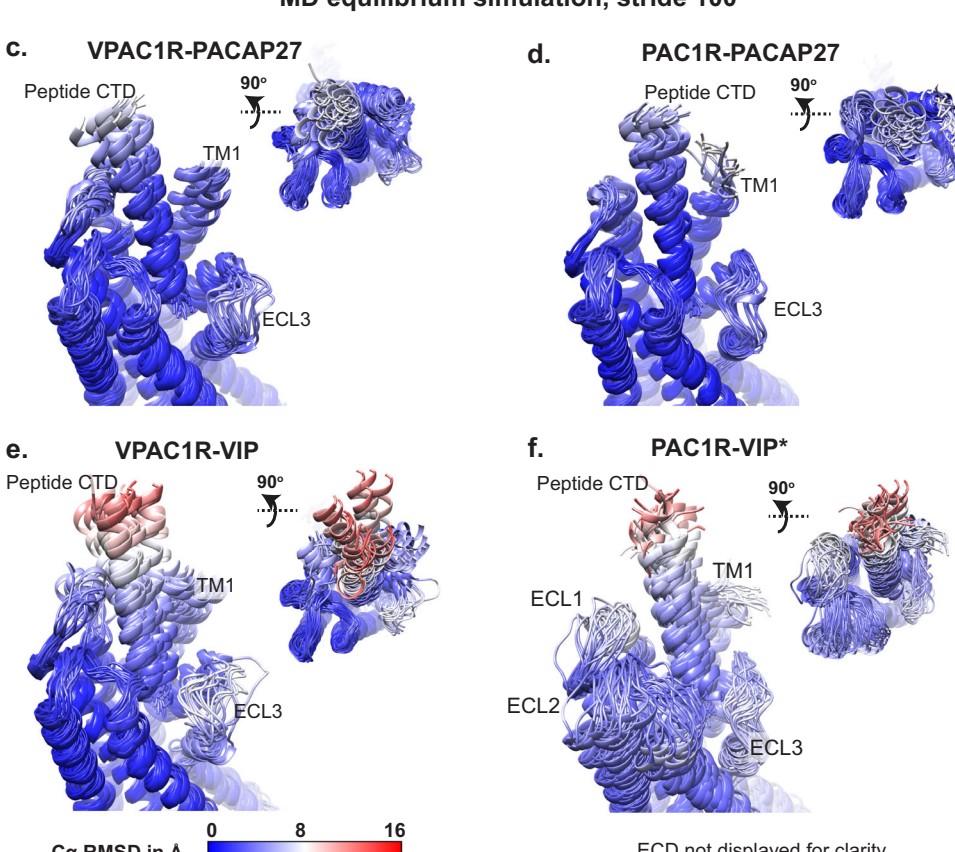

**Fig. 3 | Dynamic analyses of receptor-peptide complexes by cryo-EM 3D variability analysis and MD simulations. a, b** VPAC1R-VIP (green), VPAC1R-PACAP27 (blue) and PAC1R-PACAP27 (pink) atomic coordinate frames derived from the cryoSPARC 3DVA extreme frames (frame 0 and 19) from each component (component 0 to 2) were overlayed and different frames were coloured in a colour gradient according to their peptide-receptor pair (Comp 0, frame 0 lowest saturation, Comp 2, frame 19 highest saturation). **c–f** Snapshots of the MD equilibrium simulation from the experimental structures of VPAC1R-PACAP27, PAC1R-PACAP27 and VPAC1R-VIP, as well as the PAC1R-VIP homology model (as indicated by asterisk \*). Simulations were run without the ECD-ECL1 disulfide bond present. Snapshots were extracted from the simulations as PDBs (extracting every 100th frame), opened in Chimera and coloured by Cα RMSD in Å, in a colour scale from 0 Å = blue to 16 Å = red. The receptor and peptide backbones are displayed in ribbon format and shown as front view of the receptor TMs and peptide and top view. ECD residues of the PAC1R and VPAC1R receptor are not displayed for clarity. For the PAC1R-VIP homology model, the MD simulations did not consider the full-length Gs protein (only the Gα H5 was retained).

reduced ranges of motion during the MDSs, including a more closed and stable ECL3 conformation that was similar for both receptors (Fig. 3c, d). While the conformational space sampled in the cryo-EM 3DVA was more limited than the MDSs, there was a good correlation with respect to the extent and types of observed dynamics (Fig. 3a–e).

Details of the occupancies of receptor-peptide interactions extracted from the MD equilibrium analyses are summarized in Supplementary Table 3 (non-polar contacts) and Supplementary Table 4 (H-bonds). While some of the conserved interactions observed in the consensus cryo-EM structures were reflected in the MDSs, e.g. >40%

occupancy for D3$^{PACAP/VIP}$-R2.60 and T7$^{PACAP/VIP}$–K2.67, most interactions were receptor-dependent, rather than peptide-dependent, and were mainly independent of sequence conservation within the peptides. In the VPAC1R structures, the H-bond interaction between S2$^{PACAP/VIP}$ and E7.42 had the highest frequency (>90%), however, the equivalent interaction was less persistent in the PACAP27-PAC1R structure (S2$^{PACAP}$-E7.42; 5.4% occupancy) (Supplementary Table 4). In the PAC1R structure, D3$^{PACAP}$ exhibited high occupancy H-bond interactions with both R2.60$^{PAC1R}$ (92.4%) and Y1.47$^{PAC1R}$ (56.3%), whereas in VPAC1R structures, only R2.60$^{VPAC1R}$ interactions with D3$^{PACAP}$ (73.8%) or D3$^{VIP}$ (43%) were observed (Supplementary Table 4).

Non-polar interactions of A4$^{VIP}$ and V5$^{VIP}$ were restricted to ECL2 (I289$^{ECL2}$) and the proximal segment of TM5 (W5.36) for VPAC1R, whereas the longer I5$^{PACAP}$ side chain also formed persistent interactions with TM7 (L7.39$^{PAC1R}$/M7.39$^{VPAC1R}$) and ECL3 (E374$^{PAC1R}$) (Supplementary Table 3, Supplementary Fig. 8), which could support the more closed ECL3 conformation seen in both the cryo-EM and MD data with PACAP27 bound. Further comparison of the persistence of contacts predicted for VIP-VPAC1R, and PACAP27-VPAC1R, indicated that the largest differences in interaction patterns occurred in the most flexible regions of the complex, in particular, the ECD, ECLs and top of the TM1/stalk (Supplementary Fig. 9a, b). PACAP27 contacts with ECL1 and ECL2 residues, as well as W5.36$^{VPAC1R}$ of TM5, were more prolonged than for interaction of these regions with VIP. In contrast VIP formed more contacts with ECL3/TM7 residues and the top of TM1. Interestingly, VIP also had a stronger engagement with residues at the far N-terminus helix 1 of the ECD (Helix1$^{ECD}$) (Y39$^{ECD}$/I43$^{ECD}$), and these interactions were distinct from those formed by Helix1$^{ECD}$ with PACAP27, which preferentially interacted with E36$^{ECD}$ and V40$^{ECD}$ (Supplementary Table 3, Supplementary Fig. 9a, b).

Overall, VIP exhibited less prolonged H-bonded interactions with VPAC1R over the time course of the MDSs, particularly for the mid-peptide residues, when compared with PACAP27, which displayed more frequent and prolonged H-bonds with the PAC1R and VPAC1R (Supplementary Table 4); this aligns with fewer H-bonds observed in the consensus VIP-VPAC1R structure compared to the PACAP27 bound VPAC1R and PAC1R structures. Moreover, there were no H-bonds with an occupancy of >50% of the simulation frames in VIP-VPAC1R, except for the aforementioned S2$^{VIP}$ and E7.42$^{VPAC1R}$ pair (Supplementary Table 4). In contrast, PACAP27 S11$^{PACAP}$ and Y13$^{PACAP}$ interacted with ECL2 (D287$^{VPAC1R}$, 82.5%; D298$^{PAC1R}$, 65.8%) and TM1 (T1.33$^{VPAC1R}$, 50.6%; D1.33$^{PAC1R}$, 86.6%), respectively. Similarly, while R12$^{VIP}$, R14$^{VIP}$ and K15$^{VIP}$ displayed limited H-bond interactions with VPAC1R, the equivalent residues in PACAP27 had extensive engagement with both VPAC1R (R12-E36$^{ECD}$, 61.5%; R14-E204$^{ECL1}$, 49.3%; K15-D287$^{ECL2}$, 44.2%) and PAC1R (R12-D301$^{ECL2}$, 59.7%; R14-D215$^{ECL1}$, 97.6%; K15-D298$^{ECL2}$, 38%). Interestingly, both VIP and PACAP27 had more persistent H-bond interactions for C-terminal polar residues (K20, K21, Y22) at the VPAC1R compared with PACAP27 at the PAC1R in the MD simulations (Supplementary Table 4).

## Modelling VIP interaction dynamics at PAC1R

To further interrogate potential differences in peptide-receptor interactions that could contribute to peptide selectivity at the PAC1R, we ran equilibrium MD on a homology model of VIP bound to PAC1R that was generated using the PACAP27-PAC1R complex structure as the template (Fig. 3f, Supplementary Table 3, 4). VIP exhibited stark differences in the engagement with the PAC1R receptor (Fig. 3f), relative to simulations of VIP bound to the VPAC1R and PACAP27 bound to both receptors. The peptide N- and C-termini interactions with the PAC1R were more dynamic than in the PACAP27-bound VPAC1R and PAC1R structures. Even more striking were the larger ranges of motions for ECL1$^{PAC1R}$ and ECL2$^{PAC1R}$ as well as the TMs (in particular TM4 and TM5), indicating generally higher dynamics in the entire receptor-peptide complex relative to the other simulations

(Fig. 3f). Nonetheless, while the overall motion in the VIP-PAC1R MD was greater than observed for all the other complexes, ECL3 had a relatively closed and more stable conformation, when compared to the VIP-VPAC1R MDS, which was more open and dynamic (Fig. 3f vs 3e).

Despite starting from the deep binding pose of the template PACAP27-PAC1R complex, VIP-PAC1R exhibited fewer stable interactions for the N-terminal peptide residues with the TM bundle both with respect to polar and non-polar interactions, in comparison to all other peptide-receptor complexes (Supplementary Table 3, 4, Supplementary Fig. 9c, d). This was particularly true for the engagement of receptor residues deep within the TM bundle that contribute to receptor activation, e.g. D3$^{VIP/PACA27}$ - R2.60$^{VPAC1R/PAC1R}$, residues that are also conserved in other class B1 receptors[14]. The transient nature of these interactions is also in-line with the high conformational variance of the complex over the course of the simulation (Fig. 3f). While neither PACAP27 or VIP exhibited persistent H-bonding of the peptide C-terminal residues with PAC1R (Supplementary Table 4), the non-polar interactions formed with PACAP27 were more extensive and prolonged (Supplementary Table 3). However, there were more persistent H-bonds formed between basic residues in the mid-region of VIP and PAC1R (R12, R14), relative to VIP-VPAC1R (Supplementary Table 4).

As noted above, the high-resolution PACAP27-bound structures enabled modelling of water networks around the N-terminal peptide residues that highlighted the differential distribution of water molecules below S2$^{PACAP27}$ for VPAC1R and PAC1R structures (Supplementary Fig. 5). As such, we also analysed water-mediated polar interactions between receptors and the N-terminal peptide residues during the MD (Supplementary Table 5). While overall, the frequency of water-mediated H-bonds was similar across the receptor complexes, for both peptides, there were clear receptor-dependent differences in the engagement of S2$^{VIP/PACAP27}$ that primarily formed direct interactions with VPAC1R and water-mediated interactions at the PAC1R (Supplementary Tables 4, 5). Thus, while MD revealed weaker engagement of the VIP N-terminus with PAC1R, this did not appear to depend on differences in water-mediated interactions.

## Partial binding/unbinding simulations

In addition to equilibrium MD, we probed peptide partial binding and unbinding events for each peptide N-terminus with the TM core (ECLs and TMD) of each receptor using supervised MD (SuMD) and meta-dynamics (summary in Supplementary Table 6), to gain insight into whether the interactions formed during peptide engagement and disengagement with the receptor might also contribute to differences in VIP selectivity for the VPAC1R over the PAC1R, which are not evident for the PACAP27 (Supplementary Movie 3). The quantitative data from these simulations are summarized in Supplementary Table 7 (partial binding and unbinding generic contacts) and Supplementary Table 8 (partial binding and unbinding hydrogen bonds), and visual summaries are illustrated in Fig. 4 and Supplementary Figs. 10, 11. Binding energies of the peptides to the receptors are summarized in Supplementary Table 9 and show that the PACAP27 peptide, when bound to the VPAC1R or PAC1R, has similar binding energies, whereas the VIP-PAC1R complex has less negative energy, which is in line with the structural and pharmacology data in this paper, showing that the lower affinity peptide VIP forms a less stable complex with PAC1R. Per-residue contributions to the binding energies are also summarized in Supplementary Fig. 12, in which charged residues of the mid-region of the peptide (R12, R14) show strong negative binding energies, particularly for the PAC1R bound peptides.

Overall, the unbinding events during the simulations followed similar interaction patterns for all peptides (Supplementary Tables 7, 8, Fig. 4, Supplementary Movie 3). During unbinding, R14, K15 and K21 in the mid-region of the peptides formed the most persistent interactions with the TM core, particularly with polar residues in ECL1 and

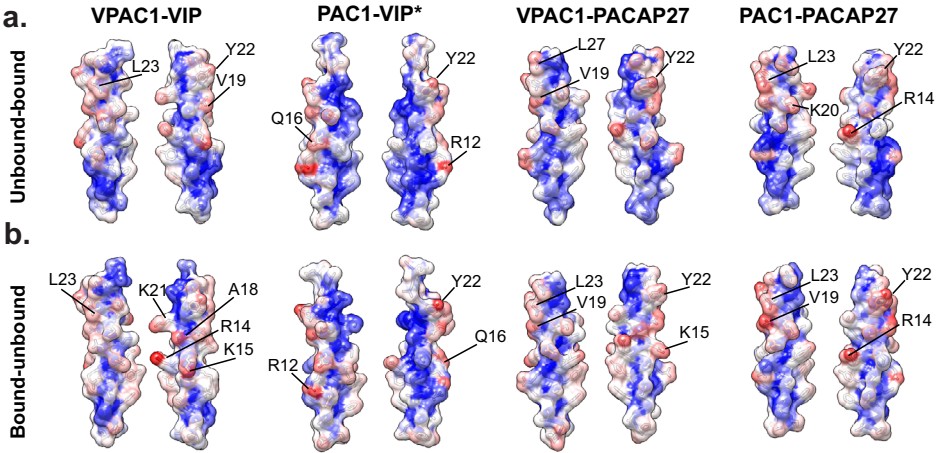

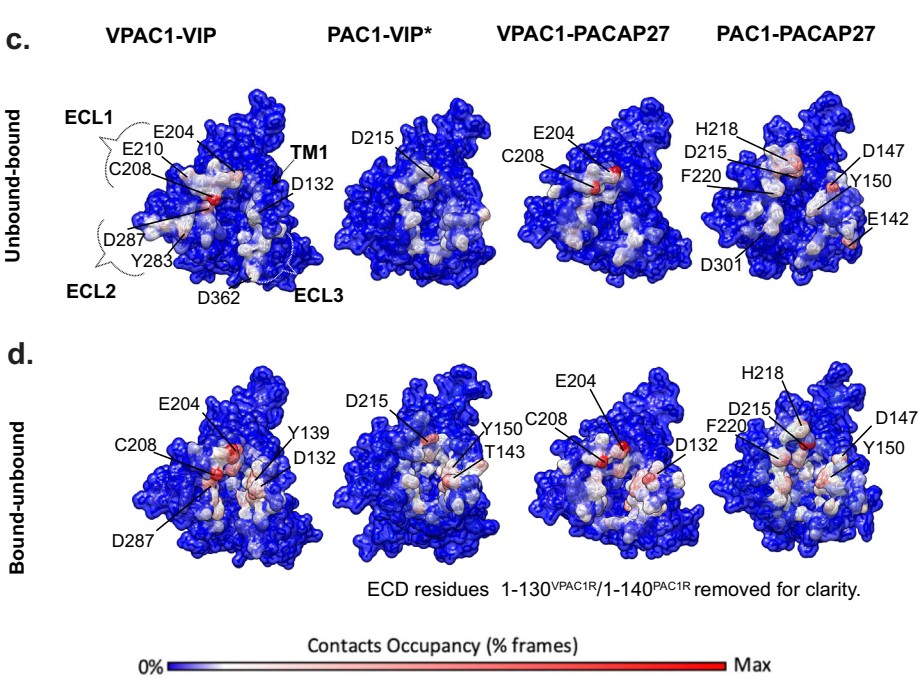

**Fig. 4 | Contact occupancies based on unbinding and binding MD simulations comparing contacts between VIP and PACAP27, and PAC1R and VPAC1R.** MD simulations were performed on the experimental PAC1R-PACAP27, VPAC1R-VIP and VPAC1R-PACAP27 complexes and the PAC1R-VIP (homology model, as indicated by asterisk *). The total occupancy (% MD frames) for each atom is plotted on the surface of the peptide (**a**, **b**) or receptor (**c**, **d**) according to a colour scale from 0 % contacts occupancy = blue to maximum contacts occupancy = red. **a**, **b** Two-sided view of VIP and PACAP27 peptide (surface representation) with occupancies plotted based on the binding simulation (**a**) or unbinding simulation (**b**). **c**–**d** Top view of receptors PAC1R and VPAC1R (surface representation) with occupancies plotted based on the binding simulation (**c**) or unbinding simulation (**d**). ECD residues of PAC1R and VPAC1R were removed for clarity.

ECL2, however, fewer and more transient interactions of VIP with PAC1R were observed during the unbinding process. R14$^{PACAP27}$ formed stronger polar interactions with both VPAC1R and PAC1R than VIP formed with these receptors, while the interactions of K15$^{VIP}$ with ECL2 were stronger for the VPAC1R relative to PAC1R. In addition, while K21$^{PACAP}$ H-bond interactions with ECL1 were observed during the binding and unbinding simulations of PACAP27 to both receptors, K21$^{VIP}$-ECL1 H-bonds were more persistent during the binding and unbinding simulations from the VPAC1R, relative to the PAC1R (Supplementary Table 8). Indeed, H bond interactions of K21$^{VIP}$ with PAC1R ECL1 were not observed for unbinding simulations, and only a very low

occupancy interaction with D215$^{PAC1R\ ECL1}$ was observed in the PAC1R binding simulations.

Of the N-terminal residues, F6$^{VIP/PACAP}$ had the most persistent interactions for all complexes during unbinding, but these were more stable for VIP at both VPAC1R and PAC1R relative to F6$^{PACAP27}$ (Supplementary Table 7). The most stable H-bond interactions involving N-terminal peptide residues during the unbinding simulation were those between S2$^{VIP/PACAP27}$ and E7.42$^{VPAC1R}$, or D3$^{VIP/PACAP27}$ with residues in TM7$^{PAC1R}$ and TM2$^{PAC1R}$, further supporting the differential contributions of these residues to interactions with the two receptors (Supplementary Table 8). Interestingly, mutations of VIP revealed

changing Ser2 to Ala did not have a major effect on VIP binding in VPAC1R-expressing CHO cells[23], whereas alanine mutation of residues 1 (His), 3 (Asp), and 6 (Phe) revealed these residues are important for VIP and PACAP27 peptide binding to both the VPAC1R and PAC1R [23–25].

Consistent with the observations for peptide unbinding, the initial binding of the peptide to the receptor core predominantly involved charged residues, with basic amino acids in the mid-region of the peptide engaging with ECL1, often in concert with interactions at the top of TM1, particularly for the PAC1R (Supplementary Movie 3, Fig. 4, Supplementary Tables 7, 8). For the VPAC1R, the most persistent interactions were with E204[ECL1], C208[ECL1] and E210[ECL1], while for PAC1R they included Q214[ECL1], D215[ECL1] and H218[ECL1] and polar residues at the top of TM1 (Y1.25[PAC1R], E1.28[PAC1R], D1.33[PAC1R] and Y1.36[PAC1R]) (Fig. 4c, Supplementary Fig. 11). These interactions facilitated subsequent engagement with ECL2/ECL3. A unique interaction, even though with only low occupancies, is D2.68 of VPAC1R with R14 of VIP (Supplementary Table 8), whereas R14 engages with TM1 rather than with TM2 in the other complexes. An alanine mutant of D2.68 (D196A) in VPAC1R resulted in a 500-fold reduction in EC$_{50}$ of VIP-mediated cAMP production consistent with an important role of this side chain in VIP-mediated VPAC1R activation [26].

Early interactions of the far N-terminus of each of the peptides with the TM core were also observed, however, these were not stable and within the timeframe of the simulations did not lead to productive engagement that enabled the peptides to achieve the deep interactions with the receptor core that are associated with the fully active state. Nonetheless, the secondary engagement of the peptide N-terminus with ECL2/ECL3 appeared to coordinate interactions that preceded the movement of the far N-terminal residues deeper into the binding pocket. Unique residues in ECL2 that differ between the VPAC1R and PAC1R are important for peptide-mediated receptor activation as alanine mutation of M299[PAC1R]/D301[PAC1R], which formed transient interactions with PACAP27 residues 1–5 during both the peptide binding and unbinding simulations (Supplementary Table 7), reduced PACAP38-mediated cAMP production[11]. I289A[VPAC1R] also shows reduced cAMP signalling with PACAP27[10], and this residue is also likely important for VIP-mediated cAMP production.

Interestingly, we observed kinking of the peptides for some simulations that were required to facilitate interactions of the peptides with ECL2/ECL3. In addition, in all the PACAP27 simulations, the unwinding of the far N-terminal helix was observed, enabling interactions of the peptide N-terminus with deeper residues in both the PAC1R and VPAC1R TM bundle (Supplementary Movie 3, Supplementary Fig. 10a, b). This occurred around the glycine at residue 4 of PACAP that destabilizes the helix and facilitates the unwinding. In contrast, the VIP peptide N-termini exhibit less unwinding of the helical structure in both binding and unbinding simulations (Supplementary Fig. 10). In place of glycine, VIP contains an alanine at position 4 (A4[VIP]), which is more rigid and likely contributes to the maintenance of the helix. This potentially creates a higher energy barrier for peptide binding to the active state and may contribute to peptide selectivity with VIP having a higher affinity for the VPAC1R, in contrast to PACAP27 which has a similar affinity for VPAC1R and PAC1R. To assess this, we generated chimeric peptides swapping the residue at position 4 between PACAP27 and VIP to generate Gly4 VIP and Ala4 PACAP27 and assessed the ability of these peptides to activate VPAC1R and PAC1R mediated cAMP signalling (Fig. 5a, b). At the VPAC1R these peptides had a similar potency and maximal response relative to the parent peptides. However, at the PAC1R the cAMP potency for Ala4 PACAP27 was significantly reduced relative to PACAP27 (5-fold), and Gly4 VIP was significantly enhanced (3-fold) relative to VIP. We also assessed these peptides in a PAC1R NanoBRET competition binding assay using AF568-conjugated PACAP27 as the fluorescent probe (Fig. 5c). These data revealed a 30-fold reduction in the pIC50 for Ala4 PACAP27 binding to the PAC1R relative to PACAP27, albeit there were

no significant differences between the pIC50s of Gly4-VIP and VIP. While these data are consistent with an important role of Gly4 in the ability of PACAP peptides to bind and activate both receptors, there are clearly other interactions that are also important for the selectivity of these peptides, which were revealed by the binding/unbinding simulations described above.

While the N-termini of the peptides partially unwind, in particular for PACAP27, the peptide C-termini remained helical for the largest proportion of the simulations, except for VIP in the binding/unbinding simulations to PAC1R (Supplementary Fig. 10d, h). The weak interactions of the VIP peptide with the PAC1R ECD and ECLs might be due to a lack of peptide stability for initial receptor engagement. Several studies highlight the importance of C-terminal peptide residues and their helical secondary structure for the interaction with the receptor[27,28]. Poor stability of the VIP C-terminus may therefore compromise the ability of the peptide to efficiently recognize and activate the PAC1R.

## Potential ECD-ECL1 disulfide bonds contribute to differential peptide potency

Resolution of ECL1 was relatively low in all experimental structures of the VPAC receptor family with limited ability to assign correct residue side chain rotamers. Nonetheless, in all of our cryo-EM maps there is close proximity of a pair of cysteines, in H1[ECD] and ECL1 (C25[ECD]-C219[ECL1] for PAC1R, and C37[ECD]-C208[ECL1] for VPAC1R), with the clearest density observed in the cryo-EM map for the PACAP27-PAC1R-Gs complex (Fig. 6a–c). These observed densities support the presence of a disulfide interaction in at least a proportion of the particles that contribute to the maps. To assess the importance of this putative disulfide bond in peptide mediated signalling, the two cysteines in each receptor were individually mutated to alanine (Fig. 6d, e, Supplementary Fig. 13), C25A and C219A in PAC1R and C37A and C208A in VPAC1R, and these were assessed in cAMP accumulation assays (Supplementary Fig. 13a–f). A double alanine mutant was also assessed for the PAC1R (C25A/C219A). For all mutants, there was equivalent cell surface expression to the corresponding wild-type receptors (Supplementary Fig. 13g). Mutation of either cysteine in the VPAC1R and PAC1R, significantly reduced the potency of VIP (25–200 fold) (Fig. 6d, e), and this was consistent with mutation of both cysteines for the PAC1R (Fig. 6e). In contrast, the single mutations at either receptor resulted in a much smaller reduction in potency for PACAP38 and PACAP27, which was only significant for PACAP27 at C37A VPAC1R or C25A PAC1R relative to the respective wildtype receptor (Fig. 6d, e). Moreover, the reduced potency of PACAP27 was restored in the double mutant of the PAC1R, which was not significantly different from wildtype PAC1R (Fig. 6e). A partial restoration of function for double mutation over single mutation of cysteines involved in disulfide bond formation has been observed in other systems, including the PTH1R, where serine mutations of the two cysteines (C281[ECL1/TM3] and C351[ECL2]) that form a conserved disulfide bond between ECL1/TM3 and ECL2 in all class B1 GPCRs, was less deleterious for ligand binding compared to the single mutants [29].

To probe how the presence of a disulfide bond between the ECD and ECL1 might alter peptide interactions, we performed additional MD equilibrium analyses of receptor-peptide complexes in the presence of the disulfide bond, and compared these to the previous equilibrium MD where this bond was absent (Supplementary Fig. 14, Supplementary Tables 3, 4). Modelling of the disulfide bond in the VPAC1R decreased the ECD mobility, and stabilized ECL3 and the top of TM1, particularly when bound to VIP (Supplementary Fig. 14a). Interestingly, when bound to PACAP27, inclusion of the disulfide also led to increased mobility of ECL1 and the distal half of the ECD that links to TM1 (Supplementary Fig. 14b). Overall, the persistence of interactions with VPAC1R and the contact binding energies were similar for both peptides with or without modelling of the disulfide bond, with some minor differences in the interactions of basic residues

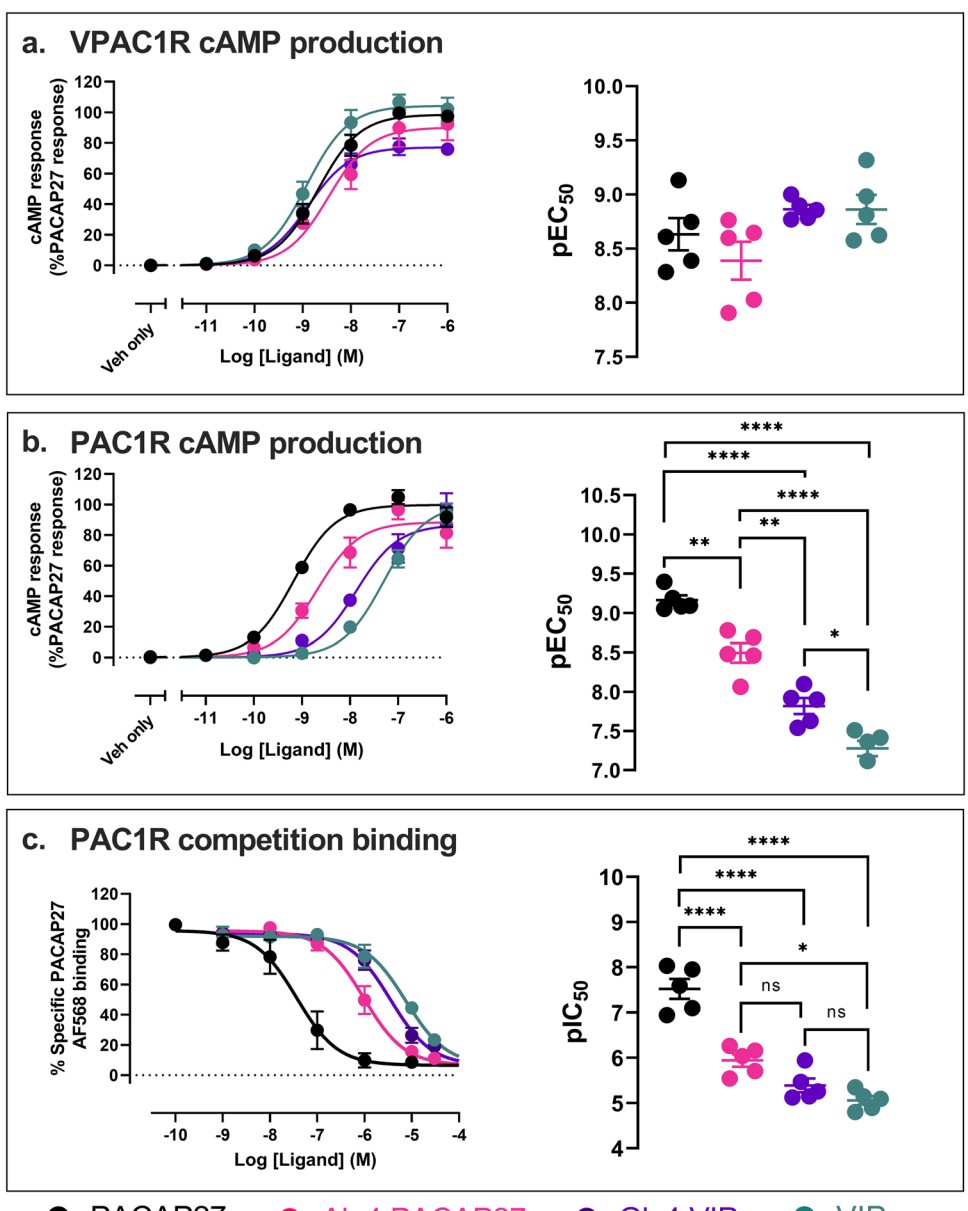

**Fig. 5 | cAMP production and competition binding assays using PACAP27, VIP and chimeric peptides Ala4-PACAP27 and Gly4-VIP.** Pharmacological characterization of PACAP27 (black), VIP (turquoise) and the chimeric peptides Ala4-PACAP27 (pink) and Gly4-VIP (purple); (**a**) VPAC1R cAMP production, (**b**) PAC1R cAMP production and (**c**) PAC1R competition binding. **a**, **b** Left, Concentration–response curves for cAMP production normalized to % PACAP27 response (data are the mean ± SEM from 5 independent experiments); right, pEC$_{50}$ determined from each individual experiment are shown with mean ± SEM of the 5 independent experiments also shown. **c** Left, Competition binding measured using nanoBRET with AF568-conjugated PACAP27 probe (30 nM) and Nluc-PAC1R in the absence and presence of competing peptides. Data are normalized to % specific PACAP27-AF568 nanoBRET signal (data are the mean ± SEM from 5 independent experiments); right, pIC$_{50}$ values determined from each individual experiment are displayed with mean ± SEM of the data shown. pEC$_{50}$ and pIC$_{50}$ data analysed using one-way ANOVA, Tukey's post-hoc (ns = not significant, *$p < 0.05$, **$p < 0.01$, ****$<0.0001$). Source data are provided in the Source Data file.

in the mid-region of the peptides that were more evident for VIP (R12, R14, K15) (Supplementary Fig. 12).

For the PAC1R, there was a general increase in ECD mobility when the disulfide was modelled, especially at the C-terminal end of ECD Helix 1, and this was particularly true for the PACAP27 bound receptor (Supplementary Fig. 14d). For both VIP and PACAP27 peptides, there was decreased mobility of the PAC1Rn ECD loop; however, there were peptide-specific differences in the mobility of the PAC1Rn loop in the presence of the disulfide, with increased mobility in the PACAP27 bound complex, yet decreased mobility in the VIP-bound complex. For this latter complex, there were also decreases in the mobility of the far N-terminal residues of the ECD and ECL1, though this was not observed

for the PACAP27-bound receptor, which had increased mobility induced by the presence of the disulfide bond (Supplementary Fig. 14c vs 14d). However, for both peptide complexes, there was generally decreased dynamics of ECL2, TM4 and TM5 that extended through to the intracellular half of TM4. In general, the interaction pattern was similar for PACAP27-PAC1R regardless of the presence or absence of the disulfide bond, although changes in the interaction pattern of H1$^{PACAP27}$ and reduced stability of interactions of R14$^{PACAP27}$ with ECL1-$^{PAC1R}$ were evident (Supplementary Tables 3, 4). By far the greatest impact of the presence of the disulfide was on the predicted interactions of VIP with the PAC1R, where increased interactions were observed between the far N-terminal residues and the receptor,

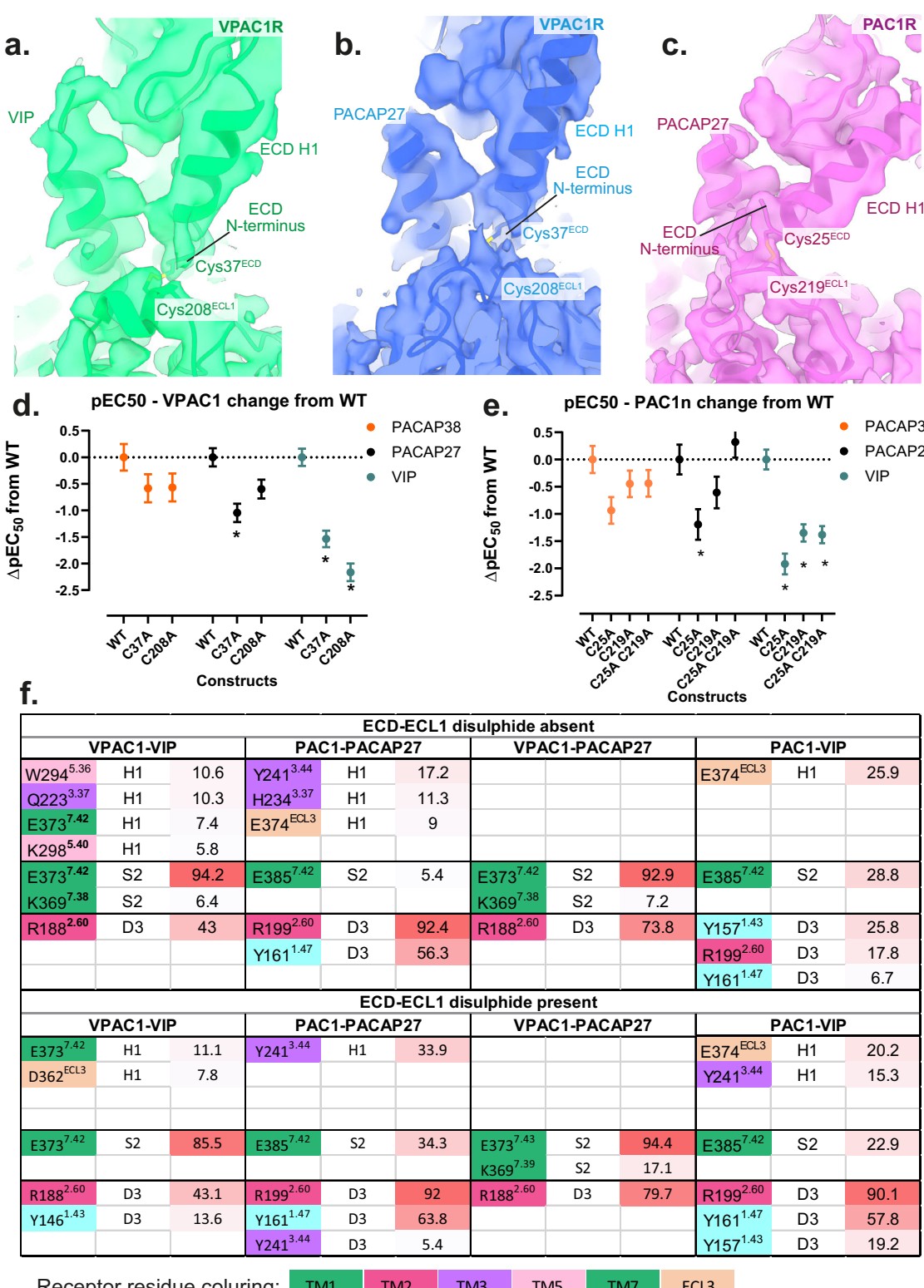

including a dramatic increase in the stability of the D3$^{VIP}$ H-bond with R2.60$^{PAC1R}$ (Supplementary Table 4). There were also increased frequencies of occupancy of contacts between T7$^{VIP}$, T11$^{VIP}$ and K15$^{VIP}$ with ECL1/ECL2 and the tops of the proximal TMs, along with a general increase in the stability of interactions between the peptide and the PAC1R ECD. Calculation of peptide binding energies (Supplementary Table 9) also show more negative energy for the VIP-PAC1R complex in the presence of the disulfide bond, suggesting that the ECD-ECL1 disulfide stabilizes the VIP-bound complex.

Collectively, our data suggest the potential for ECD-ECL1 disulfide bond formation in VPAC family receptors that can modify receptor function in a ligand-dependent manner. Nonetheless, the cryo-EM data indicate that this proposed ECD-ECL1 bond is not ubiquitously present, at least under the conditions used for expression and purification for structure determination, despite the high level of conservation of the relevant cysteines from VPAC1R and PAC1R from different species (Supplementary Fig. 15). Future studies assessing the redox state of the bond for receptors expressed in different cellular backgrounds are

**Fig. 6 | Potential disulfide bond structure and function in the ECD helix 1 and ECL1 of VPAC1R and PAC1R. a**–**c** Cryo-EM maps of the receptor-alone focused refinements of VPAC1R-VIP (green), VPAC1R-PACAP27 (blue) and PAC1R-PACAP27 (pink) shown as side view with a zoom onto the ECD Helix 1 and ECL1. Maps are shown in transparent and model backbones are shown in ribbon format. The relevant cysteines are shown as stick models, with the putative disulfide bond present. Due to ambiguity, the disulfide bond was not modelled in the final, deposited models, however, the relevant cysteine residues are in a position to potentially form a bond. Cysteines of interest are labelled according to their residue number: Cys37 (ECD) and Cys208 (ECL1) of the VPAC1R and Cys25 (ECD) and Cys219 (ECL1) of the PAC1R. **d**–**e** Differences in cAMP potency (pEC50) derived from concentration response analysis (data in Supplementary Fig. 13 comparing wild-type receptors to alanine mutant receptors lacking the relevant cysteines); **d** pEC50 of WT and C37A and C208A VPAC1R single mutations in response to PACAP38 (orange), PACAP27 (black) and VIP (dark cyan); e: pEC50 of WT, C25A, C219A and C25A/C219A in response to PACAP38 (blue), PACAP27 (red) and VIP (green). The data for panels d and e are the mean ± SEM, $n = 3$ independent experiments. Differences in potency between WT and mutant receptors was assessed using a one-way ANOVA and Dunnett's post-test. Statistically Significant changes ($p < 0.05$) are indicated with asterisk (*). **f** Tabular comparison of N-terminal peptide residues (H1, S2, D3) hydrogen bonds with receptor residues, comparing the presence and absence of the ECD-ECL1 disulfide bond during the MD equilibrium simulation from the experimental structures of VPAC1R-PACAP27, PAC1R-PACAP27 and VPAC1R-VIP, and the PAC1R-VIP homology model. Interactions are listed by percentage of occupancy of frames, with the background colouring representing maximum occupancy % (red) and minimum occupancy % (white). Receptor residue background colouring is according to their residue numbering (TM transmembrane helix, ECL3 extracellular loop 3). Table is based on data presented in Supplementary Table 4. Source data for panels d and e are provided in the Source Data file.

## G protein engagement with receptors

All investigated peptide-bound PAC1R and VPAC1R structures exhibited conserved features of active state class B1 GPCRs, in particular the opening of the base of TM5 and TM6 to enable binding of the C-terminal Gs αH5 helix into the receptor TM bundle (Fig. 1b, Supplementary Fig. 16), facilitated by the unwinding of the top of TM6 to induce a kink around the centre of the helix. There was also overall high agreement in the conserved residue interactions of each receptor and the Gs heterotrimer. This includes conserved H-bond interactions extending from Gαs residues Q384$^{G.H5.16}$, R385$^{G.H5.17}$, and E392$^{G.H5.24}$ (Supplementary Fig. 16, Table 2). Interactions in the cryo-EM structures were in agreement with the MD equilibrium simulations, with the highest occupancy H-bonds between D381$^{Gs}$ and Q384$^{Gs}$ with K$^{5.64}$ of both receptors and L394$^{Gs}$ with R$^{6.37}$ of both receptors (Supplementary Table 10). Non-polar contacts with the highest occupancies in the MD simulations were also similar in all structures, for example H387$^{Gs}$ with L$^{3.57}$, as well as L393$^{Gs}$ with S$^{6.41}$ (Supplementary Table 10).

Despite the overall high similarities in Gs binding, an overlay of the (receptor-aligned) cryo-EM consensus structures show an offset of the PAC1R bound G protein, with shift of the Gs αN, Gβ and Gγ, in comparison to the VPAC1R bound G protein (Fig. 1b). This difference is, at least in part, also sampled in the 3DVA frames (Supplementary Fig. 7, Supplementary Movie 1) and therefore the consensus structures represent only a snapshot from the dynamic continuum of the G protein interactions.

## Discussion

Based on our collective results, we propose a model for peptide selectivity at PAC1 and VPAC1 receptors. Peptide binding to class B1 peptide GPCRs involves a complex series of sequential and dynamic interactions that enables initial peptide engagement with the receptor extracellular face to facilitate peptide N-terminal engagement deeper within the TM binding cavity where it can engage residues that are required to initiate conformational transitions necessary for receptor activation. In all cases, the stability of the interactions between the VIP or PACAP27 C-terminus and receptor ECD were important for the persistence of peptide interactions with the receptor core, and for interactions in the intermediate binding state including polar interactions of ECL1/TM1 with peptide mid-region basic amino acids, and engagement of the N-terminal activation domain with the ECLs. Moreover, for both VIP and PACAP27, the binding simulations indicated that the peptides kink to facilitate interactions that support transition of the peptide N-terminus to deeper binding. This peptide flexibility mirrors observations for the related glucagon-like peptide 1 receptor (GLP-1R) where GLP-1 analogues become kinked during presumed binding/unbinding dynamics observed in cryo-EM data

suggesting that peptide, as well as receptor conformational dynamics, contribute to binding/unbinding paths of class B1 peptide agonists[30,31].

Overall, VIP makes fewer and less stable contacts with the receptor ECD in the experimentally derived structure and/or models of interactions with active VPAC1R and PAC1R. This in turn leads to weaker interactions between basic residues in the mid-region of the peptide and acidic/polar residues within ECL1 and the top of TM1, relative to PACAP peptides. Nonetheless, these interactions with the ECLs and TM1 are key metastable intermediate interactions of the peptide N-terminus with the receptor core during binding, and are consistent with previous photoaffinity labelling and alanine mutagenesis studies where key identified TM1 residues are important for VIP affinity at the VPAC1R[32]. In addition, the necessity of stable peptide interactions with the ECD and ECLs to facilitate N-terminal peptide binding in the TM bundle was consistent with another MD study investigating binding of VIP/PACAP and peptide mutants to the PAC1R[33]. Studies also identified that replacing the receptor N-terminus of the secretin receptor (SecR) with that of VPAC1R could generate a similar level of VIP-mediated cAMP production as wild-type VPAC1R[34], however, interestingly replacing the PAC1R ECD with VPAC1R was not sufficient to generate a similar level of VIP-mediated cAMP production[35]. However, a chimeric receptor with the PAC1R TMs 2−7 and VPAC1R ECD and TM1 was sufficient to generate a more potent VIP response, akin to that of the VPAC1R[35], highlighting the importance of residues within the VPAC1R ECD and TM1 residues for high-affinity VIP binding, and implicating these domains as key drivers for the lower affinity of VIP for the PAC1R.

The stability of C-terminal peptide interactions with the ECD is a key factor for the selectivity of PAC1R for PACAP peptides relative to VIP. In addition, the conformational dynamics exhibited by the ECD/ peptide C-termini impacted the stability of intermediate binding interactions of the mid-region of the peptides with the ECLs and top of the TM bundle. This enables productive initial engagement of the peptide N-terminus with ECL3 and ECL2, prior to deep binding within the TM bundle. Modelling of VIP-PAC1R binding revealed that VIP had fewer stable contacts with the ECD than the other peptide-receptor complexes. As such, instability is transmitted to VIP's mid- and N-terminal interactions, including the key peptide activation residues, H1-S2-D3. The predicted unwinding of the VIP C-terminus observed in the PAC1R partial unbinding and binding simulation may further reduce the efficiency of initial binding of the VIP peptide to the PAC1R ECD and ECLs[36,37].

The ability of the VIP or PACAP27 peptides to engage deep within the receptor core is also dependent upon conformational flexibility of both the PAC1R ECL3 and the far N-terminus of the peptides. ECL3 in the PAC1R is less dynamic than the corresponding ECL3 in the VPAC1R and the transition to deep binding in the case of PACAP27 is facilitated by destablization of the N-terminal helix induced by G4$^{PACAP}$ and unwinding of the peptide N-terminus that allows the N-terminal

residues greater flexibility to form new interactions. In contrast, the VIP N-terminal helix is more stable and rigid, which in the context of the less dynamic PAC1R ECL3 would be expected to increase the energy barrier for VIP to engage deep within the TM bundle to activate the receptor, relative to VIP binding to the more open and dynamic VPAC1R TM bundle. Previous studies using NMR are consistent with this, showing that PACAP27 can form an N-terminal beta-turn, for which Gly4 is essential, whereas Ala4 at the equivalent position in VIP would not be able to form the same structure[38]. The higher plasticity of ECL3 in the VPAC1R can support the formation of productive interactions with key N-terminal residues of both the flexible PACAP27 and the stable VIP N-terminal helix that guide the deeper engagement, and thus accounting for the high affinity of both peptides for the VPAC1R. Our observations from structural and MD data are supported by pharmacological data that revealed the importance of glycine at position 4 for high affinity and high cAMP potency at PAC1R, which was not required for the VPAC1R. Consistent with our experimental data, previous studies also showed that Ala4 PACAP27 had 30-fold lower affinity at the PAC1R relative to PACAP27, and Gly4 VIP exhibited 10-fold higher affinity for PAC1R than the parent peptide when assessed in rat brain membranes[24]. Moreover, in addition to position 4, substitution at position 5 of the PACAP27 peptide to the equivalent residue in VIP also reduced PAC1R affinity, suggesting that both positions contribute to PAC1R peptide affinity[24]. Overall, these data are consistent with an important role for flexibility in the far N-terminus of peptide ligands for PAC1R affinity, and thus N-terminal rigidity for the selectivity of VIP for the VPAC1R. The importance of N-terminal flexibility for ligand affinity to class B1 GPCRs has been noted previously for analogues of exendin-4, a GLP-1R agonist that natively has a glycine at position 4. Here the substitution of glycine for L-alanine (that stabilizes helical conformation) reduced the affinity of the peptide by 8-fold and cAMP potency by 100-fold[31]. In contrast, a D-Ala substitution, which destabilizes α-helical structure, had similar cAMP potency to exendin-4. Future structural studies on VIP and/or PACAP analogues bound to PAC1R will likely assist in understanding these mechanisms. Nonetheless, our data emphasise the importance of analysing dynamics for studying receptor-peptide interactions and assessing the stability of non-conserved and conserved interactions.

Interestingly ECL3 residues, in particular the second half of the ECL3 and top of TM7 region, have relatively low sequence similarity in class B1 GPCRs. Receptor chimera constructs of VPAC1R and PAC1R identified VPAC1R-TM6/ECL3 as important for VIP recognition and VIP-mediated receptor activation[35]. Alanine substitution of acidic residues at the TM6/ECL3 boundary of the GLP-1R (D372$^{6.61}$,E373$^{6.62}$) reduced GLP-1 potency for cAMP production[17], and switching R380$^{7.35}$ at the GLP-1R ECL3/TM7 with the equivalent residue of VPAC1R also reduced GLP-1 affinity[39]. In the equivalent position as R380$^{GLP1R}$, VPAC1R has a proline residue, PAC1R has a lysine and SecR has a glutamic acid/methionine. The low conservation of ECL3 residues might indeed represent the location of peptide selectivity or underline the need for differences in dynamics of ECL3 by different receptors to allow peptide binding with residues of select receptors.

Regardless of ECL3 flexibility, to facilitate stable binding deep in the pocket of VPAC1R and PAC1R, regardless of the bound peptide, conserved key interactions of H1, D3 and F6 with residues of the TM bundle are necessary. Mutations of these residues to alanine reduce binding of VIP to VPAC1R, however surprisingly the conserved S2 is less important[19]. Differences were observed in water molecule networks in the vicinity of S2 and D3 of the PAC1R vs VPAC1R, where the S2$^{PACAP27}$ side chain penetrates deeper into the PAC1R TM bundle, relative to VPAC1R where it occupies the position of a water molecule that is present in the VPAC1R binding pocket. Therefore, it is possible that structural water molecules are capable of replacing some of the interactions of peptide residues, which might be an important consideration in the development of small molecule agonists. Indeed, a

comparison of water networks in structures of the related GLP-1R with bound peptide and small-molecule agonists revealed conserved structural waters[17], and in the case of the small molecule agonist PF-06882961, an extensive water-mediated hydrogen bond network deep in the TM bundle that fills the pocket to replace interactions occupied by peptide N-terminal side chains in peptide-bound GLP-1R structures (Supplementary Fig. 5).

The potential novel disulfide bond between ECL1 and the far N-terminal Helix 1$^{ECD}$ may also contribute to peptide selectivity at the VPAC family of receptors. Our mutagenesis data demonstrated that the inability to form this bond markedly, and selectively, decreased VIP potency, with equilibrium MD linking this to changes in the conformational dynamics of VIP, reduced persistence of interactions with ECL1/TM1 and subsequent engagement of peptide N-terminal interactions deeper in the TMD. Thus, the presence of the disulfide bond in VPAC1R and PAC1R may increase the probability of the formation of productive interactions of the peptide N-terminus that would aid in VIP engagement deeper in the core of the receptor. Earlier studies on class B1 GPCRs that have equivalent cysteines proposed these formed a disulfide bond, but there is contradictory data in the literature regarding the existence of this ECD-ECL1 disulfide bond, and its functional importance[40,41]. It is likely that the redox state of the receptor will be dependent upon cell background and that this will be influenced by the physiological or pathological state of the host tissue.

Collectively, our work provides molecular insight into key interactions for activation of VPAC1R and PAC1R by endogenous peptide agonists. Moreover, the work reveals distinct conformational dynamics in a receptor and peptide-dependent manner that underpins the differential peptide selectivity of these receptors.

## Methods
### Peptides
PACAP38, PACAP27 and VIP were purchased from ChinaPeptides (Shanghai, China). Ala4 PACAP27 and Gly4 VIP were purchased from GL Biochem Ltd. (Shanghai, China). These peptides were diluted in 0.05% acetic acid and stored in aliquots at 100 μM, taking into account peptide content and purity.

### Receptor constructs for mammalian cell transfection
An HA signal peptide was inserted in place of the native signal peptide of the receptors and a FLAG epitope was incorporated into the receptor N-terminus immediately after the signal peptide. A His tag was fused to C-terminus of the receptor. 3C protease cleavage sites were inserted between both the FLAG and His tags and the receptor. These modifications were assessed in PAC1R WT and VPAC1R WT and did not alter receptor pharmacology ([9], Supplementary Fig. 1a). The following PAC1R and VPAC1R mutants were also generated using Quikchange mutagenesis: PAC1R C25A, PAC1R C219A, PAC1R C25A/C219A, VPAC1R C37A and VPAC1R C208A. For all PAC1R constructs, we used the PAC1R null (PAC1n) splice variant. A construct containing N-terminal NLuc immediately after the native signal peptide of PAC1R was also generated in pcDNA3.1. The insertion of the luciferase did not alter the receptor pharmacology PAC1 (Supplementary Fig. 17).

### Mammalian cell culture and transfection
COS-7 cells were used for the chimeric peptide and cysteine mutagenesis pharmacological assays due to their lack of endogenous expression of PAC1R and receptor activity-modifying proteins, reported to potentially heterodimerize with PAC1R and VPAC1R[42]. Cells were cultured in Dulbecco's Modified Eagle's Medium (DMEM) (Invitrogen) supplemented with 5% v/v heat-inactivated FBS at 37˚C and 95% $O_2$ /5% $CO_2$ in a humidified incubator. COS-7 cells were plated in 10 cm dishes at approximately 1 million cells/dish. Cells were transfected 24 h later, with 5 μg of either WT or mutant PAC1R or VPAC1R receptor DNA. DNA and PEI (in a 1:6 ratio) were each diluted in 150 mM NaCl, and then

combined and incubated for 10 min, before the mixture was added to the 10 cm dish. The next day, cells were harvested from the 10 cm dishes using Trypsin-Versene (PBS + 0.5 mM EDTA, pH 7.4), and were seeded at a density of 15,000 cells/well into clear 96-well culture plates and incubated overnight at 37 °C in 5% $CO_2$ for cAMP accumulation assay the next day.

Chinese Hamster Ovary (CHO) cells were used for the VPAC1R pharmacological structural construct validation assays. CHO cells were transfected in suspension using receptor DNA (100 ng/well) and PEI (600 ng/well) and seeded at a density of 30,000 cells/well into clear 96-well culture for cAMP accumulation assay.

### Whole cell competition assays

**AF568-PACAP27 probe generation.** The pituitary adenylate cyclase-activating peptide 27 (PACAP27, Native fragment) contains 27 amino acid residues. The structure-activity data on PACAP peptides suggest that position 21 is not important for binding to the receptor and thus suitable for fluorophore attachment. Therefore, the lysine residue at position 21 was replaced with a cysteine residue ($^{21}$C) for the facile conjugation of fluorophore via thiol-maleimide click reaction. The resulting target peptide $^{21}$C-PACAP27 (Mutant fragment) with amidated C-terminus was synthesized by standard Fmoc-solid phase peptide synthesis method[43] and purified by Reversed-phase high-performance liquid chromatography (RP-HPLC). The NHS ester (succinimidyl ester) of Alexa Fluor™ 568F was purchased from ThermoFisher Scientific (Catalog number: A20003). The thiol-maleimide conjugation reaction was carried out by our published protocol[44]. The purity of the resulting probe, *AF568-PACAP27* ($^{21}$C(Alexa568)PACAP27), determined by analytical RP-HPLC was over 99%. The molecular weight of the probe was confirmed by Matrix-Assisted Laser Desorption/Ionization Time-of-Flight Mass Spectrometry (MALDI-TOF MS).

**NanoBRET binding assay.** COS-7 cells were plated in 10 cm dishes at approximately 1 million cells/dish. Cells were transfected 24 h later, using 3 μg of Nluc-PAC1R DNA using the method described above and then seeded at a density of 15,000 cells per well in white-bottom 96-well culturplates 24 h prior to assay. On the day of the assay, cell culture media was replaced with BRET buffer (0.1% ovalbumin, 10 mM HEPES, 1x HBSS, pH 7.45) and incubated at 4 °C for two hours. A final concentration of 30 nM of the fluorescent probe (AF568-PACAP27) and increasing concentrations of competing ligand (PACAP27, VIP, Ala4-PACAP27 or Gly4-VIP) were added simultaneously and incubated in the dark at 4 °C. After 1.5 h, furimazine (10 μM final concentration) was added to each well and the plate incubated for a further 30 min in the dark at 4 °C before reading. Filtered light emissions were measured at 460±40 nm (donor channel) and at 610-LP nm (long pass) (acceptor channel) using a PHERAstar plate reader (BMG Labtech). The raw NanoBRET ratio was calculated by dividing the luminescence from the acceptor channel by the donor channel. Data were expressed as % AF568-PACAP27 BRET signal in the absence of competing ligand.

### Cyclic AMP accumulation assays

On the day of assay, growth media was replaced with stimulation buffer (phenol-free DMEM containing 0.1% (w/v) OVA and 0.5 mM 3-isobutyl-1-methylxanthine, pH 7.4) and incubated for 30 min at 37 °C in 5% $CO_2$ before cells were stimulated with increasing concentrations of agonist (PACAP38, PACAP27 or VIP for the cysteine mutagenesis pharmacological assays and PACAP27, VIP, Gly4-VIP or Ala4-PACAP27 for the chimeric peptide pharmacological assays). The reaction was terminated after 30 min by aspiration of the buffer and addition of 50 μl of ice-cold ethanol. Upon evaporation of ethanol, 75 μl of lysis buffer (5 mM HEPES, 0.1% (w/v) OVA, 0.3% (w/v) Tween20, pH 7.4) was added to the cells. cAMP detection was performed using a LANCE cAMP Detection Kit (PerkinElmer) for the construct validation and cysteine mutagenesis pharmacological assays and cAMP Gs HiRange

kit (Cisbio) for the chimeric peptide pharmacological assays. The plates were read using an EnVision plate reader with excitation at 320 nm and emission at 615 nm[45]. All values were converted to an absolute concentration of cAMP using a cAMP standard curve performed in parallel and normalized to PACAP27/forskolin and vehicle controls.

### Antibody staining and FACS analysis

To confirm cell surface receptor expression of FLAG-tagged wild-type and mutant receptors, cells were assessed for anti-FLAG antibody labelling by FACS[13]. Cells were washed with 1 × PBS, harvested with Versene, then pelleted by centrifugation at 350 g, 4 °C for 5 min. Samples were then blocked with 5% (w/v) BSA in FACS buffer (1 × HBSS, 10 mM HEPES, 0.1% (w/v) BSA, pH 7.4) for 30 min on ice. After blocking, anti-FLAG M2 mouse antibody (MIPS TC13-03-01-03) in FACS buffer (2 μg/mL final concentration) was added and incubated for 1 h on ice, followed by three washes with FACS buffer. Goat anti-mouse AF647 (Invitrogen #A21235) in FACS buffer (1 μg/mL final concentration) was then added and incubated for 1 h on ice and in the dark. After a further 3 washes, SYTOX Blue nucleic acid stain was added (1:2000) 5 min before reading. Cells were then read on the BDFACS Canto II flow cytometer (BD Biosciences) with the following lasers: 405 nm (SYTOX blue) and 633 nm (for AF647 detection). FlowJo software v10 was used to analyse the data, gating to whole cells (through FSC-H and SSC-H) and live cells (through negative SYTOX blue stain) within the whole cell population. Mean AF647 of the live cell population was used as an indicator of cell surface receptor expression.

### Quantification and statistical analysis

The pharmacological data in figures and tables are reported as mean + standard error of the mean (SEM) with the number of biological replicates indicated in the figure and table legends where "n" represents the number of biological replicates performed. Concentration-response data were analyzed in GraphPad Prism version 9.4.0 using a three-parameter logistic curve to derive $pEC_{50}$ values for the cyclic AMP accumulation. Competition binding data were analyzed in GraphPad Prism using the one site−Fit log $IC_{50}$ model to derive $pIC_{50}$ values. For the cysteine mutagenesis data, comparisons of multiple different groups were performed using one-way analysis of variance (ANOVA), followed by Dunnett's post-hoc test with the control group being the WT receptor, with significance accepted at $p < 0.05$. For the chimeric peptide data, comparisons of multiple different groups were performed using one-way ANOVA, followed by Tukey's post-hoc test, with significance accepted at $p < 0.05$.

### Constructs for protein expression and purification

Receptor constructs of PAC1R (Liang et al, Mol Cell 2020) and VPAC1R were modified to include purification tags at the N-terminus (FLAG tag) and C-terminus (HIS tag), 3C protease cleavage sites (downstream of the FLAG tag and upstream of the HIS tag) and replacing the original signal peptide with the hemagglutinin (HA) signal peptide for enhanced expression. For the PAC1R, the splice variant PAC1null (PAC1n) was used[9]. A dominant negative form of human Gαs (DNGs[46]) was used together with human His$_6$-tagged Gβ$_1$ and Gγ$_2$. For the PAC1R-PACAP27-Gs complex, an earlier DNGs version (DNGsV1) was used (missing the A366 mutation, as utilized in ref. [47]). Both VPAC1R-Gs complexes used DNGsV2 (containing A366S mutation)[46]. All constructs were prepared in baculovirus expression vectors for complex generation.

C-terminally His-tagged Nanobody 35 was used to stabilize the complex for structural studies[48], which was expressed and purified using previously described protocols[31]. The Nanobody 35 construct was transformed into *Escherichia coli* (*E. coli*) BL21DE3 cells. Colonies were grown in Terrific broth (TB) at 37 °C and expression was induced using 1 mM IPTG when optical density ($OD_{600}$) at ~0.6 was reached, and

cells were harvested after over-night incubation at room temperature. Pelleted cells were lysed in ice-cold buffer (containing 200 mM Tris pH 8.0, 0.5 mM EDTA, 500 mM sucrose, 2.5 mg/l leupeptin, 160 mg/l benzamidine, 50 µg/ml lysozyme), and, after incubation, debris was removed using centrifugation. The His-tagged Nanobody 35 (Nb35) was purified using Ni-NTA resin in affinity chromatography and eluted using 20 mM HEPES, 100 mM NaCl and 200 mM imidazole, pH 7.5. Eluate was flash-frozen in concentrations around 2 mg/ml and stored at −80 °C until use.

### Insect cell expression

Cell cultures of *Trichoplusia ni* (*Tni*) insect cells (Expression systems) were used for the expression of all complexes. Using viral titres optimized for protein expression, receptor, DNG$\alpha_s$, G$\beta_1$ and G$\gamma_2$ were co-expressed using the baculovirus system, with insect cells at 3 million (VPAC1R-PACAP27 complex) or 4 million (PAC1R-PACAP27 and VPAC1R-VIP complex) cells per ml cell count infected with the separate baculoviruses, prior to incubation at 27 °C for 48 h, harvesting and storing the cell pellets at −80 °C.

### Complex purification

Cell pellets were thawed in 30 mM HEPES pH 7.4, 50 mM NaCl, 2 mM MgCl$_2$, 5 mM CaCl$_2$ supplemented with complete Protease Inhibitor Cocktail tablets (Roche) and benzonase nuclease (Merck Millipore). Complex formation was initiated by the addition of 10 µM peptide (China Peptides) and incubation for 20 min at room temperature. After the addition of Nb35–His (20 µg/mL) and apyrase (25 mU/mL, NEB); the suspension was incubated for another 20–30 min at room temperature. The complexes were solubilized using 0.5% (w/v) lauryl maltose neopentyl glycol (LMNG) and 0.03% (w/v) cholesterol hemisuccinate (CHS) (Anatrace) for 1 h at 4 °C. The solubilized complexes were batch-bound to M1 anti-Flag affinity resin for 2 h at room temperature. The resin was packed into a glass column and washed with 40 column volumes of 20 mM HEPES pH 7.4, 100 mM NaCl, 2 mM MgCl$_2$, 5 mM CaCl$_2$, 1 µM peptide, 0.01% (w/v) LMNG and 0.0006% (w/v) CHS before bound material was eluted in buffer containing 10 mM EGTA and 0.2 mg/mL FLAG peptide. The complex was then concentrated using an Amicon Ultra Centrifugal Filter (MWCO 100 kDa) in the presence of 0.1 mM TCEP (for the VPAC1R-PACAP27 sample only), and subjected to size-exclusion chromatography on a Superdex 200 Increase 10/300 column (GE Healthcare) that was pre-equilibrated with 20 mM HEPES pH 7.4, 100 mM NaCl, 2 mM MgCl$_2$, 1 µM peptide, 0.01% (w/v) LMNG and 0.0006% (w/v) CHS (for the VPAC1R-PACAP27 sample only, 0.1 mM TCEP was added to separate complex from contaminating aggregates). Eluted fractions consisting of receptor and G protein complex were pooled and concentrated. For the VPAC1R-VIP complex, a further purification step was conducted after 3 C cleavage of receptor tags and overnight incubation with Talon resin. After washing and elution of the complex containing 300 mM imidazole, eluate was concentrated and subjected again to size-exclusion chromatography. Final complex samples were concentrated to 3.9 mg/ml (VPAC1R-VIP) 5.15 mg/ml (PAC1R-PACAP27) and 7.9 mg/ml (VPAC1R-PACAP27), flash-frozen in liquid nitrogen and stored at −80 °C. Purity and stability of the complex following thawing was confirmed by SDS-PAGE and negative-stain transmission electron microscopy.

### SDS-PAGE and negative-stain TEM analysis

Samples collected from size-exclusion chromatography were analyzed by SDS−PAGE to assess sample quality. For SDS−PAGE, precast gradient TGX gels (Bio-Rad) were used. Gels were stained by Instant Blue (Expedeon). For negative-stain TEM, aliquots of flash-frozen protein complex were diluted to ~0.002 mg/ml in detergent-free buffer prior to applying 4 µl of sample onto a glow-discharged Cu grid with carbon film and subsequently negatively stained in uranyl formate droplets. Grids were imaged on a Talos 120 C at 120 keV and 1.96 Å/pixel for the

VPAC1R-PACAP27 sample and Tecnai T12 TEM at 120 keV and 2.06 Å/pixel for the VPAC1R-VIP and PAC1R-PACAP27 samples. Particles were picked and 2D classified using Relion 3.1[49].

### Preparation of vitrified specimen

Acetone pre-washed electron microscopy grids (Ultrafoil R1.2/1.3 Au 300 mesh) were glow-discharged and 3 µL of the sample was applied to the grid in a Vitrobot Mark IV chamber (Thermo Fisher Scientific), set to 100% humidity at 4 °C. The sample was blotted for 10 s with a blot force of 19 and then flash frozen in liquid ethane.

### Data acquisition

Data were collected on a Titan Krios microscope (Thermo Fisher Scientific, Waltham, MA, USA) operated at an accelerating voltage of 300 kV with a 50 µm C2 aperture. Data collection details and differences between samples are summarized in Table 1. A Gatan K3 direct electron detector, positioned post a Gatan Quantum energy filter (Gatan, Pleasanton, CA, USA) was used to collect movies as compressed TIFFs in normal-resolution mode. Beam-image shift was used to acquire data from 9 surrounding holes and one image per hole for VPAC1R-VIP and PAC1R-PACAP27 samples, or 4 images per holes for the VPAC1R-PACAP27 sample, after which the stage was moved to the next collection area using a custom SerialEM script [50].

### Data processing

For the VPAC1R-VIP dataset, two subsets were initially processed separately due to changes in total dose during the data collection (see Table 1) and particles were combined for the final 3D reconstructions. Collected movies were subjected to motion correction using MotionCor2[51] and CTF estimation was performed using the Gctf software[52] on non-dose-weighted micrographs, implemented in Relion v3.1-beta/v3.1.0[49]. The particles were picked using the automated procedure in crYOLO[53] with a GPCR-trained model as initial picking weights and coordinates were imported into Relion. Subsequent data processing steps were carried out using Relion v3.1-beta/v3.1.0[49]. Particles were extracted initially using a box of 64 pixels, and after curation of particles in 2D and 3D classifications, re-extracted using a final box size of 288 for the PAC1R-PACAP27 and VPAC1R-VIP data (pixel size of 0.83 Å/pixel) and final box size of 320 for the VPAC1R-PACAP27 data (pixel size of 0.65 Å/pixel). As an initial 3D reference, a previous GPCR complex map (based on the PAC1R-PACAP38 dataset) was used and 60 Å low-pass filtered to prevent model bias. 3D references in subsequent steps were derived from the data itself. Subsequent rounds of 3D classifications, 3D refinements and 3D classification without angular and translational alignment, were used to create a homogenous set of particles which was further subjected to Bayesian particle polishing and CTF refinements (as implemented in Relion v3.1-beta/v3.1.0) and a final global 3D refinement. In post-processing, different masks were applied on the global refinement map, resulting in global resolutions (FSC = 0.143) of 2.3 Å using the global, wide mask and 2.4 Å using a tight mask (excluding parts of the extra-cellular domain of the receptor, α-helical domain of the Gα protein and micelle) for the PAC1R-PACAP27 and VPAC1R-PACAP27 data, and 2.7 (tight) and 2.9 (wide) Å for the VPAC1R-VIP data (Supplementary Fig. 2, Table 1). To better resolve features of the receptor loops and extracellular domain, the global refinement map was subjected to 3D classification and 3D refinement with fine angular sampling using a mask including only the receptor and peptide (referred to as 'receptor-only' refinement). The receptor-only map resulted in a global resolution (FSC = 0.143) of 2.5 Å for the PAC1R-PACAP27 and VPAC1R-PACAP27 data and 3.0 Å for the VPAC1R-VIP data (Supplementary Fig. 2, Table 1). Local resolution estimates and maps were produced in Relion (Supplementary Fig. 2d–f). Differently B-factor sharpened maps were created for modelling of the higher resolution areas of the TMs and waters (Relion automated B-factor sharpening) or lower resolution areas

(Relion manual B-factor sharpening of −25 to −20). The refinement maps, the auto B-factor sharpened, post-processed consensus maps (used for modelling of the TM bundle and waters), as well as the receptor-only focused maps are deposited as additional maps. All masks were created with a custom script using e2proc3d.py from EMAN2[54].

## Atomic model refinement

Atomic coordinates were refined into the PAC1R cryo-EM maps based on the previously published PACAP38 model (PDB 6P9Y), with mutating/removing peptide residues and Gs residues to account for the correct versions used in each complex. A homology model of VPAC1R was initially created in SwissModel[55], based on a template search of the receptor amino acid sequence, and was used as a starting model for refinements into the cryo-EM maps of the VPAC1R complexes. The initial model was refined into the global maps initially by flexible fitting using Isolde[56], and subsequent rounds of Real-Space refinement using Phenix[57] and manual inspection in Coot[58]. After creating a complete consensus model, the majority of the model (Gα, Gβ, Gγ, TMs and receptor core waters) was further refined into the higher-resolution, auto-sharpened, post-processed map using Phenix and Coot. Lower resolution areas (in particular ECD, ECL1, ICL1−3 were further refined using the receptor-only maps, with variable sharpening using Coot (Blur/Sharpen maps) and Isolde/ChimeraX[59] and refined in Phenix and Coot. Molprobity scores[60] as well as the Phenix Cryo-EM Comprehensive Validation report were used throughout as quality control of the model geometries and are reported in Table 1. Due to model ambiguity, residues were omitted or stubbed (sidechains were removed) from the models after final refinements and prior to deposition. ECD residues in proximity to the peptide were retained. For both VPAC1R structures (receptor chains R), residues of the ECD 51−66, 73−84, 94−100, 106−117 (except Cys in conserved disulfide bonds) were stubbed for final deposition. For the PAC1R structure (receptor chain R), residues 88−113 (PAC1Rnull ECD loop) and 139−145 (ECD/TM1 stalk) were deleted and residues 40−57, 65−76, 114−130 (PAC1R ECD, except Cys in conserved disulfide bonds) were stubbed for final deposition. The disulfide bonds between ECD Helix 1 and ECL1 were removed in the final PDB model, but Cys residues were retained. For the G proteins, amino acids from the α-helical domain (62−204) and missing loops of the Gα protein (253−260) have been omitted in all models for deposition. In addition, both VPAC1R models are missing residues 301−307 (chain A, Gα subunit), and VPAC-VIP is additionally missing residues 163−166 (chain B, Gβ subunit).

## Non-experimental, homology model building of the PAC1R-VIP complex

A homology model of PAC1R-VIP was created using our experimental PAC1R-PACAP27 model as a template, with removing the PACAP27 peptide chain and replacing it with the VIP peptide chain from our experimental VPAC1R-VIP structure. These coordinates were used as a starting model for refinements into the cryo-EM map of the PAC1R-PACAP27 complex. The model was further refined and assessed (protein geometries and clash scores) using Phenix and manual inspection in Coot. This non-experimental model was used in molecular dynamics experiments.

## Model residue interaction analysis and general structure analysis tools

Interactions between chains were analyzed using the "Dimplot" module within the Ligplot+ program (v2.2) ([61], Supplementary Fig. 4). Hydrogen bonds were also assessed using the UCSF ChimeraX package, with relaxed distance and angle criteria (0.4 Å and 20-degree tolerance, respectively; Table 2, Fig. 2, Supplementary Figs. 5, 16). Visualization of structures and production of images was performed using the UCSF Chimera package (v1.14)[62] from the Computer Graphics

Laboratory, University of California, San Francisco (supported by NIH P41 RR-01081) and ChimeraX (support from National Institutes of Health R01-GM129325). For comparison of complex structures, unless otherwise stated, complexes were aligned by receptor chains using the *matchmaker* command in Chimera/ChimeraX.

## 3D variability analysis and non-uniform refinement in Cryosparc

For the data refinement and analysis in Cryosparc, the Relion particle stacks from the global consensus refinement as well as the consensus refinement map (as reference volume) were imported into the cryosparc v2 pipeline[63]. A consensus refinement in Cryosparc using the Homogeneous refinement tool was produced, which was used as an input for the 3D variability analysis[64]. For the variability analysis, the wide mask created automatically during refinement in Cryosparc (including micelle) was applied. The frames of the 3 principal components generated in the 3D variability analysis were visualized using the ChimeraX volume series tool and shown in the recorded movies. Backbone models of the Cryosparc 3D variability frames were modelled into the extreme frames (#0 and #19) of each component using Isolde, based on the final consensus model. The Cryosparc non-uniform refinement tool[65] was used as alternative refinement method, to test resolvability of lower resolution areas, using default parameters and particle stacks imported from the Relion consensus refinement. The output maps were used to validate the Relion-derived maps and models and deposited as additional maps.

## Systems preparation for Molecular dynamics (MD) simulations

Missing residues (mainly in the receptors and Gαs) were added as described elsewhere[14]. The full-length models of the PAC1R:PACAP27:Gs:Nb35, VPAC1R:VIP:Gs:Nb35, and VPAC1R:PACAP27:Gs:Nb35 complexes were prepared both in the presence and absence of the potential disulfide bond between C37ECD and C208ECL1 (VPAC1R) or C25ECD-C219ECL1 (PAC1R), in six distinct simulation systems, with the CHARMM36[66] force field using VMD[67] and in-house python HTMD[68] and TCL (Tool Command Language) scripts. Pdb2pqr[69] and propka[70] software were used to add hydrogen atoms appropriate for a simulated pH of 7.0; the protonation of titratable side chains was checked by visual inspection. The structures were superimposed on the PAC1R (PDB ID 6P9Y) from the OPM database[71] to orient the receptor prior to insertion in a rectangular pre-built 126 Å × 116 Å 1-palmitoyl-2-oleyl-sn-glycerol-3-phosphocholine (POPC) bilayer; lipid molecules overlapping the receptor were removed. TIP3P water molecules were added to the 126 Å × 116 Å × 180 Å simulation box using the VMD Solvate plugin 1.5 (Solvate Plugin, Version 1.5. at http://www.ks.uiuc.edu/Research/vmd/plugins/solvate/). Overall charge neutrality was maintained by adding Na+ and Cl- counter ions to a final ionic concentration of 150 mM using the VMD Autoionize plugin 1.3 (Autoionize Plugin, Version 1.3. at http://www.ks.uiuc.edu/Research/vmd/plugins/autoionize/).

For the MD simulations of the homology-modelled PAC1R:VIP complex, the Gs subunits Gβ, Gγ, and Gα (excepted helix H5, residues 371−394) were removed along with Nb35. The resulting ternary complex (PAC1R-VIP-Gs(H5)) was prepared for MD simulations as reported above, in the presence and absence of the potential disulfide bond between C25ECD-C219ECL1. The simulations using the PAC1R-VIP homology model are indicated by asterisk * throughout the manuscript figures.

## Systems equilibration and MD settings

ACEMD[72] was used for both equilibration and MD productive simulations. Isothermal-isobaric conditions (Langevin thermostat[73] with a target temperature of 300 K and damping of 1 ps−1 and Berendsen barostat[74] with a target pressure of 1 atm) were employed to equilibrate the systems through a multi-stage procedure (integration time step of 2 fs). Initial steric clashes between lipid atoms were reduced

through 2500 conjugate-gradient minimization steps, then a 2 ns MD simulation was run with a positional constraint of 1 kcal mol$^{-1}$ Å$^{-2}$ on protein atoms and lipid phosphorus atoms. Subsequently, 20 ns of MD were performed constraining only the protein atoms and 60 ns constraining the protein backbone alpha carbons. In the final stage, all the restraints were released up to a total simulation time of 100 ns.

Productive trajectories in the canonical ensemble (NVT) at 300 K (four 500 ns-long replicas for each system, Supplementary Table 2) were computed using a thermostat damping of 0.1 ps$^{-1}$ with an integration time step of 4 fs (through hydrogen mass repartitioning[75] and the M-SHAKE algorithm[76]) to constrain the bond lengths involving hydrogen atoms. The cut-off distance for electrostatic interactions was set at 9 Å, with a switching function applied beyond 7.5 Å. Long-range Coulomb interactions were handled using the particle mesh Ewald summation method (PME)[77] by setting the mesh spacing to 1.0 Å. Trajectory frames were written every 100 ps of simulations.

Four replicas (1μs each) were computed for the fully modelled complex PAC1R:VIP:Gs(H5).

### Non-equilibrium simulations

G$_\beta$, G$_\gamma$, G$_\alpha$ (with the exception of helix H5, residues 371–394) and Nb35 were removed from the full-length models of the experimentally determined complexes PAC1R:PACAP27:Gs:Nb35, VPAC1R:VIP:Gs:Nb35, VPAC1R:PACAP27:Gs:Nb35, and the homology-modelled PAC1R:VIP:Gs:Nb35 complex (based on the PAC1R-PACAP27 structure). The resulting four systems were embedded in a pre-built 100 Å × 100 Å 1-palmitoyl-2-oleyl-sn-glycerol-3-phosphocholine (POPC) bilayer, prepared for MD simulations, and equilibrated as reported above.

The equilibrated four systems underwent the same unbinding/binding protocol (Supplementary Table 6), in analogy with[14]. Briefly, a well-tempered metadynamics[78] simulation was employed on the distance between the centroid of the residues composing the N-terminal amino acids of VIP or PACAP27 (residues 1 – 16), and the TMD of PAC1R or VPAC1R (residues 140–401 and 130–391, respectively) to dissociate the agonists from the TMD (four replicas for each system); successively three supervised MD[79,80] binding simulations were started from each unbinding trajectory (12 replicas for each system) and performed as long as the N-terminal segment of the agonists was stabilized onto the surface of PAC1R or VPAC1R. For each system, the four partial unbinding simulations and the best 4 or 5 partial binding simulations out of twelve (chosen according to the closest proximity to the experimental bound state) were submitted to further classic MD sampling (usually 30–40 classic MD simulations, each one 20ns-long, seeded from configurations extracted from the binding/unbinding transitions) as reported in[14].

The MMPBSA.py[81] script, from the AmberTools20 suite (The Amber Molecular Dynamics Package, at http://ambermd.org/), was used to compute molecular mechanics energies combined with the generalized Born and surface area continuum solvation (MM/GBSA) method, after transforming the CHARMM psf topology files to an Amber prmtop format using ParmEd (documentation at <http://parmed.github.io/ParmEd/html/index.html).

For a detailed review of the MM/GBSA end-point approach see ref. [82].

### MD video production

For producing Supplementary Movie 3, a metadynamics (partial unbinding) simulation was merged with a SuMD binding simulation started from the former, for each system. These representative simulations were used as the backbone for seeding short classic MD simulations (not shown in the movies).

### MD analyses

Atomic contacts were computed using the GetContacts analysis tool (at https://getcontacts.github.io/), with the donor-acceptor threshold distance set to 3.5 Å and the angle set to 120°. Root mean square fluctuation (RMSF) values were computed using VMD after superposition of the MD trajectories frames on the α carbon of the TM domain (residues Y138$^{1.25}$ to L403$^{7.60}$ for PAC1R, A128$^{1.25}$ to L391$^{7.60}$ for VPAC1R). The DSSP (dictionary of the secondary structure of proteins) analysis[83] was performed using AmberTools[84].

Analyses of the equilibrium simulations involving the homology-modelled PAC1RVIP:Gs(H5) complex were performed on the second half of the four 1-μs-long replicas (considered aggregate sampling time of 2 μs).

### Reporting summary

Further information on research design is available in the Nature Portfolio Reporting Summary linked to this article.

### Data availability

The structural data (atomic coordinates and cryo-EM density maps) generated in this study have been deposited in the Protein Data Bank (PDB) and Electron Microscopy Data Bank (EMDB) databases under accession codes PDB 8E3Z/EMD-27874 (VPAC1R-VIP complex); PDB 8E3Y/EMD-27873 (VPAC1R-PACAP27 complex); PDB 8E3X/EMD-27872 (PAC1R-PACAP27 complex). All other data generated in this study are provided in the Supplementary Information and Source Data files that are provided. Source data are provided with this paper.

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

## Acknowledgements

This work was supported by the National Health and Medical Research Council of Australia (NHMRC) (program grant 1150083, P.M.S./A.C.; Senior Research Fellowship 1155302, D.W.; Senior Principal Research Fellowship 1154434, P.M.S.). R.D. was supported by Takeda Science Foundation 2019 Medical Research Grant and Japan Science and Technology Agency PRESTO (18069571). C.A.R is grateful for a Royal Society Industry Fellowship. This work was supported by the Monash University Ramaciotti Centre for cryo-electron microscopy (negative staining TEM) and the Monash University MASSIVE high-performance computing facility (cryo-EM data processing). The authors would like to thank Yan Zhang, H. Eric Xu and Yi Jiang for providing access to their structural data for VPAC1R [10] prior to publication, George Christopoulos and Villy Julita for generating the expression constructs and alanine mutants of the PAC1R and VPAC1R, as well as Rachel M. Johnson and Matthew J. Belousoff for assisting with map and model refinements and validation.

## Author contributions

S.J.P., Y.-L.L. and Y.L. prepared protein complex samples. R.D. performed sample vitrification and collected cryo-EM data. SJP processed data, built the models and S.J.P., G.D., C.R., D.W analysed structures. G.D. and C.R. designed, performed and analysed the MD simulations and bioinformatics. M.M.F. validated the expression constructs in cAMP assays. J.L., P.Z. and D.W. conducted and/or analysed pharmacology assays. M.A.H generated AF568-PACAP27. S.J.P., G.D. and J.L. prepared figures and tables. S.J.P. and G.D. prepared movies. S.J.P., P.M.S and D.W. wrote initial draft of the manuscript. A.C., P.M.S., D.W. designed the project and provided financial support. All authors reviewed and edited the manuscript.

## Competing interests

P.M.S and A.C are co-founders and shareholder of Septerna Inc. D.W. is a shareholder of Septerna Inc. The remaining authors declare no competing interests.
