## [Peer Review File · Nature Communications]

Understanding VPAC receptor family peptide binding and selectivityREVIEWER COMMENTS

Reviewer #1 (Remarks to the Author):

The VIP subfamily of family B peptide-liganded GPCRs consists in three members, PAC1, VPAC1 and VPAC2. PACAP binds with high affinity to PAC1, VPAC1 and VPAC2. VIP binds with high affinity only to VPAC1 and VPAC2. Knock-out of VIP and PACAP expression in mice, pharmacological experiments in rats, and supporting pharmacological data in primate cell and tissue preparations and clinical observations in humans, implicate both peptides in distinct but complementary signaling for circadian rhythmicity; learning and memory; integration of cerebrocortical function underlying organization of enteroceptive information; inflammation in both brain and periphery; at the coronary arterial endothelium in the progression of atherosclerogenesis; cytoprotection from neurodegenerative disease and tissue ischemia; vasodilatation and vasoconstriction related to migraine; food intake (mainly for PACAP compared to VIP); and central and peripheral stress responses (in particular for PACAP and PAC1) in post-traumatic stress disorder.

Thus, it is a translational imperative that tools be developed that can act to block or mimic the actions of both VIP and PACAP, in specific pathological contexts. Overlapping specificity of VPAC receptors for VIP and PACAP is a major confounder to these efforts. The receptors are all quite similar, VIP and PACAP are highly homologous in sequence, and the mode of receptor activation seems to be highly processive. The last is a particular technical challenge. Separate binding modes of N- and C-termini appear to lead to final receptor occupancy and signaling. These are especially difficult to statically capture and understand.

Previous reports, including from the authors laboratories, have established that the C-terminal decapeptide sequence of PACAP(1-38) is dispensable for PACAP binding to any of its three receptors, so that comparison of PACAP27 and VIP28 binding to PAC1 and VPAC1/2 would be sufficient to elucidate the major features of ligand specificity. It is also established that the C-terminal domain of either VIP or PACAP27 functions as an 'affinity trap' for initial binding and positioning of ligand for receptor activation, and that the N-terminal 14-15 amino acids of PACAP or VIP function in subsequent receptor activation. Previous relevant publications referenced and considered in this report (but see also. Kobayashi et al. *Nat. Struct. Mol. Biol.* 27: 274, 2020) usually state that the structural insights provided to date will aid considerably in the development of PACAP- (or VIP)-specific agonists or antagonists. Both are very much required for continued pharmacological insight into the roles of each peptide in mammalian physiology and pathophysiology. Such tools are also required for translational attempts to identify ligand-receptor dyads (PACAP-PAC1, PACAP-VPAC1/2, VIP-VPAC1/2) for drug targeting for specific therapeutic indications, and to actualize therapeutics for these targets. Yet, a clear structural basis for differential recognition of PACAP and VIP at PAC1 and VPAC1/2 does not exist, largely because the highly flexible ECD of all three receptors has resisted a high-resolution static picture of differential binding modes for each of the several possible ligand-receptor combinations.

The present report is focused on three structural ligand-receptor-G-protein complex comparisons: between VIP-VPAC1R, PACAP27-VPAC1R and PACAP27-PAC1R. It seems to succeed at what previous investigations have only approximated: a well-defined molecular accounting for differences in binding and activation, between ligands and among receptors, which is a necessary prerequisite for rational design of selective agonists, biased ligands, and potent and selective antagonists for VIP/PACAP receptors. To do this, the authors employed high resolution (2.3-2.4 Å) cryo-EM structures for each of the three ligand-receptor pairs, and molecular dynamic simulations based upon them, with an additional simulation for VIP binding to PAC1 (a low-affinity interaction difficult to study but important to understand). The emphasis of the report is on the insight about the highly processive events of C-terminal affinity trapping leading to N-terminal positioning and receptor activation, gained from dynamic simulation deriving from high-resolution static ligand postures, that cannot be obtained with static structural representations alone, and is crucial to further gains in the field.

The technical merits of the enhanced resolution of the reported structures and associated molecular dynamic simulations that can now be accomplished, especially involving the ECD of each of the two receptors, are apparent, given the previous technical limitations overcome by the experimental approach employed here. It is worth asking whether there is i) new and cogent information here for the reader of Nature Communications non-expert in structural biology and ii) new and cogent information for the expert experimentalist that will rapidly spur further progress in the field. Does this report break new ground? In the opinion of this reviewer, it does. Notably, the fact that the selectivity of the PAC1 receptor for PACAP versus VIP involves C-terminal interactions with the ECD is established here. As it is known that VIP is fully efficacious but much less potent than PACAP27 at this receptor, this new data should drive concerted efforts to design PAC1 antagonists based on these identified ECD interaction sites. Sub-family B1 quite uniquely represents a convergence of commonality of peptide-liganded GPCR mechanism, and ligand specificity in both binding and activation. This report significantly advances understanding of both through this exemplar, via a judicious choice of ligand-receptor dyads, and combination of static and dynamic representation of ligand postures. The report is therefore recommended for its general interest to readers with diverse interests in peptide GPCR structure, function, and pharmacology.

Points of consideration:

The authors do not mention how their elucidation of structural water redistribution within both PAC1 and VPAC1 might be incorporated into peptidomimetic drug design: it would have been intriguing had they chosen to do so, especially to accommodate those involved in high-diversity virtual screening for compound binding to spatially well-defined/resolved protein targets. In addition, some circumspect discussion of concrete possibilities opened up for peptidomimetic and small-molecule pharmacological agents based on the model(s) presented here might improve the legacy prospects of the report.

Title seems somewhat uncommitted: 'Structural basis for peptide binding and selectivity in the VPAC receptor family' would not overstate the scope of the report.

Annotation/labeling within the MDS movies would be helpful for the non-specialist reader.

Figure 1B has an inherent dysymmetry (PACAP complexed with both PAC1 and VPAC1, and VIP complexed only with VPAC1. Perhaps expanding the figure to two pairwise comparisons (PACAP and PAC1 and VPAC1; PACAP and VIP and VPAC1 might make cognitive processing by the reader less demanding, even if it introduces some redundancy.

Given the high resolution of the structures obtained with the Gs-associated complexes, it might be worthwhile for the authors to offer some speculation on PAC1 versus VPAC1 coupling to Gq versus Gs, although this is perhaps a subject for another venue.

There are variant forms of the N-terminal ECD of PAC1. The authors should put their choice of PAC1n in context re: functional differences among variants.

Reviewer #2 (Remarks to the Author):

This is a solid paper, advancing our understanding on the important PACAP subfamily of proteins. Unfortunately, it is very hard to read and I recommend a major revision of the text.

Examples:

" Given the high degree of homology, there is significant interest in understanding the molecular basis for peptide selectivity between the 52 PAC1R and VPACRs". If there would be a lower degree of homology, there would be no interest in understanding the molecular basis for peptide selectivity?

The cryoEM from the literature are mentioned as "static" structure and those solved by the authors not, although they are static just the same.

“Binding to and activation of class B1 peptide GPCRs involves a complex series of sequential interactions that enables engagement with the peptide to overcome the energy barriers to activation”.

It is thermal fluctuations which allow to overcome free energy barriers, not sequential interactions.

“VIP makes fewer and less stable contacts with the receptor ECD in experimentally derived structure and/or models of interactions with active VPAC1R and PAC1R, and thus is more dynamic.” Why VIP should be necessarily more dynamic? VIP might be a much more rigid peptide than the others, even if it binds less strong to the receptor.

Reviewer #3 (Remarks to the Author):

In this paper, Wootten et al present three cryo-EM structures of VPAC receptors, VPAC1R-VIP, VPAC1R-PACAP27, and PAC1R-PACAP27 complexes. Through structural analysis, they proposed peptide ligand binding specificity. In combination with MD modeling, they showed that VIP has fewer interaction with PAC1R and VPAC1R than for PACAP27 and is more dynamic in the PAC1R-VIP complex than for the PAC1R-PACAP27 complex. The structural works are built on the previous structures of VPAC1R-VIP and PAC1R-PACAP38 complexes. The structures presented here are of better resolutions and are with water molecules, provide additional insights into peptide ligand recognition.

The most important point is the selectivity of VIP for VPAC1R over PAC1R. However, the specific residues for such selectivity between these two receptors are not clearly presented. Additional mutations that can switch specificity between VPAC1R and PAC1R should be demonstrated to fully nail down the residues in the receptors for the ligand specificity

There are only nine different residues between VIP and PACAP27. The relative contribution of each residue to the specificity for VPAC1R should be demonstrated by swapping each different residue between VIP and PACAP27.

PACAP27 and PACAP38 can bind both VPAC1R and PAC1R well. The peptide binding energy of these peptides to these receptors should be analyzed to support their binding properties.

The structures of VPAC1R-VIP and PAC1R-PACAP28 complexes have been reported previously. Detailed structure comparison between the previous structures and the current structures should be presented with RMSD between these structures for the receptor and for the peptide ligands.

The previous structure of VPAC1R-VIP complex was determined with Nanobit tethering method for stabilizing the G protein complex, where the current VPAC1R-VIP structure is not. The bound Gs heterotrimer between these two structures should be analyzed to see whether there are any differences.

RESPONSE TO REVIEWER COMMENTS

Reviewer comments are shown in black and *italics*. Our author reply to reviewer comments are labelled as “**Author reply**” and are in blue with all changes outlined in the track changes version of the manuscript file.

Reviewer #1 (Remarks to the Author):

The VIP subfamily of family B peptide-liganded GPCRs consists in three members, PAC1, VPAC1 and VPAC2. PACAP binds with high affinity to PAC1, VPAC1 and VPAC2. VIP binds with high affinity only to VPAC1 and VPAC2. Knock-out of VIP and PACAP expression in mice, pharmacological experiments in rats, and supporting pharmacological data in primate cell and tissue preparations and clinical observations in humans, implicate both peptides in distinct but complementary signaling for circadian rhythmicity; learning and memory; integration of cerebrocortical function underlying organization of enteroceptive information; inflammation in both brain and periphery; at the coronary arterial endothelium in the progression of atherosclerogenesis; cytoprotection from neurodegenerative disease and tissue ischemia; vasodilatation and vasoconstriction related to migraine; food intake (mainly for PACAP compared to VIP); and central and peripheral stress responses (in particular for PACAP and PAC1) in post-traumatic stress disorder.

Thus, it is a translational imperative that tools be developed that can act to block or mimic the actions of both VIP and PACAP, in specific pathological contexts. Overlapping specificity of VPAC receptors for VIP and PACAP is a major confounder to these efforts. The receptors are all quite similar, VIP and PACAP are highly homologous in sequence, and the mode of receptor activation seems to be highly processive. The last is a particular technical challenge. Separate binding modes of N- and C-termini appear to lead to final receptor occupancy and signaling. These are especially difficult to statically capture and understand.

Previous reports, including from the authors laboratories, have established that the C-terminal decapeptide sequence of PACAP(1-38) is dispensable for PACAP binding to any of its three receptors, so that comparison of PACAP27 and VIP28 binding to PAC1 and VPAC1/2 would be sufficient to elucidate the major features of ligand specificity. It is also established that the C-terminal domain of either VIP or PACAP27 functions as an ‘affinity trap’ for initial binding and positioning of ligand for receptor activation, and that the N-terminal 14-15 amino acids of PACAP or VIP function in subsequent receptor activation. Previous relevant publications referenced and considered in this report (but see also. Kobayashi et al. Nat. Struct. Mol. Biol. 27: 274, 2020) usually state that the structural insights provided to date will aid considerably in the development of PACAP- (or VIP)-specific agonists or antagonists. Both are very much required for continued pharmacological insight into the roles of each peptide in mammalian physiology and pathophysiology. Such tools are also required for translational attempts to identify ligand-receptor dyads (PACAP-PAC1, PACAP-VPAC1/2, VIP-VPAC1/2) for drug targeting for specific therapeutic indications, and to actualize therapeutics for these targets. Yet, a clear structural basis for differential recognition of PACAP and VIP at PAC1 and VPAC1/2 does not exist, largely because the highly flexible ECD of all three receptors has resisted a high-resolution static picture of differential binding modes for each of the several possible ligand-receptor combinations.

The present report is focused on three structural ligand-receptor-G-protein complex comparisons: between VIP-VPAC1R, PACAP27-VPAC1R and PACAP27-PAC1R. It seems

to succeed at what previous investigations have only approximated: a well-defined molecular accounting for differences in binding and activation, between ligands and among receptors, which is a necessary prerequisite for rational design of selective agonists, biased ligands, and potent and selective antagonists for VIP/PACAP receptors. To do this, the authors employed high resolution (2.3-2.4 Å) cryo-EM structures for each of the three ligand-receptor pairs, and molecular dynamic simulations based upon them, with an additional simulation for VIP binding to PAC1 (a low-affinity interaction difficult to study but important to understand). The emphasis of the report is on the insight about the highly processive events of C-terminal affinity trapping leading to N-terminal positioning and receptor activation, gained from dynamic simulation deriving from high-resolution static ligand postures, that cannot be obtained with static structural representations alone, and is crucial to further gains in the field.

The technical merits of the enhanced resolution of the reported structures and associated molecular dynamic simulations that can now be accomplished, especially involving the ECD of each of the two receptors, are apparent, given the previous technical limitations overcome by the experimental approach employed here. It is worth asking whether there is i) new and cogent information here for the reader of Nature Communications non-expert in structural biology and ii) new and cogent information for the expert experimentalist that will rapidly spur further progress in the field. Does this report break new ground? In the opinion of this reviewer, it does. Notably, the fact that the selectivity of the PAC1 receptor for PACAP versus VIP involves C-terminal interactions with the ECD is established here. As it is known that VIP is fully efficacious but much less potent than PACAP27 at this receptor, this new data should drive concerted efforts to design PAC1 antagonists based on these identified ECD interaction sites. Sub-family B1 quite uniquely represents a convergence of commonality of peptide-liganded GPCR mechanism, and ligand specificity in both binding and activation. This report significantly advances understanding of both through this exemplar, via a judicious choice of ligand-receptor dyads, and combination of static and dynamic representation of ligand postures. The report is therefore recommended for its general interest to readers with diverse interests in peptide GPCR structure, function, and pharmacology.

Author reply: We thank reviewer 1 for their thorough summary of our manuscript and the positive feedback and are grateful for their opinion that our work is breaking new ground. We have addressed all of the minor comments and points of consideration as outlined below.

*Points of consideration:
The authors do not mention how their elucidation of structural water redistribution within both PAC1 and VPAC1 might be incorporated into peptidomimetic drug design: it would have been intriguing had they chosen to do so, especially to accommodate those involved in high-diversity virtual screening for compound binding to spatially well-defined/resolved protein targets. In addition, some circumspect discussion of concrete possibilities opened up for peptidomimetic and small-molecule pharmacological agents based on the model(s) presented here might improve the legacy prospects of the report.*

Author reply: We are very grateful for the suggestions by reviewer #1 in regards to water molecules in the peptide binding pocket. However, given the difference in resolution of the different maps (e.g. VPAC1R-VIP complex at 2.7 Å vs VPAC1R-PACAP27 complex at 2.3 Å) we do not want to over-interpret our data and hence were cautious in modelling waters and

addressing the water networks in the context of small molecule binding. Class B1 GPCRs have larger solvent-exposed orthosteric sites compared to Class A GPCRs, which makes it inherently more difficult to predict water networks for small-molecule binding from peptide-bound structures. Nevertheless, experimental data (e.g. from cryo-EM maps) indicating the presence of structural waters can further guide the prediction of water networks and molecule binding in silico, and therefore we have added some comments to the manuscript that our deposited data will be helpful for future in silico predictions of small molecule binding via molecular dynamics and docking experiments. Additionally, we have added some comments (as outlined below) in regards to conserved/unique water molecule positions of the related GLP-1 receptor bound to different peptides and small molecules and added an additional figure panel to Supplementary Figure 5. Changes to the text in the discussion include the addition of the following

“Differences were observed in water molecule networks in the vicinity of S2 and D3 of the PAC1R vs VPAC1R, where the S2^{PACAP27} side chain penetrates deeper into the PAC1R TM bundle, relative to VPAC1R where it occupies the position of a water molecule that is present in the VPAC1R binding pocket. Therefore, it is possible that structural water molecules are capable of replacing some of the interactions of peptide residues, which might be an important consideration in the development of small molecule agonists. Indeed, a comparison of water networks in structures of the related GLP-1R with bound peptide and small-molecule agonists revealed conserved structural waters ([17]), and in the case of the small molecule agonist PF-06882961, an extensive water-mediated hydrogen bond network deep in the TM bundle that fills the pocket to replace interactions occupied by peptide N-terminal side chains in peptide-bound GLP-1R structures (**Supplementary Fig. 5**)”.

Title seems somewhat uncommitted: ‘Structural basis for peptide binding and selectivity in the VPAC receptor family’ would not overstate the scope of the report.

Author reply: We thank the reviewer for the title suggestion; however, we believe that the current title reflects more accurately what is shown in our current manuscript, i.e. an understanding of the peptide binding and selectivity for the VPAC receptor family.

Annotation/labeling within the MDS movies would be helpful for the non-specialist reader.

Author reply: We have added more labels to the MD unbinding/binding movie (Supplementary Movie 3) to make peptide, receptor and extracellular loops and domains clearer.

Figure 1B has an inherent dysymmetry (PACAP complexed with both PAC1 and VPAC1, and VIP complexed only with VPAC1. Perhaps expanding the figure to two pairwise comparisons (PACAP and PAC1 and VPAC1; PACAP and VIP and VPAC1 might make cognitive processing by the reader less demanding, even if it introduces some redundancy.

Author reply: We followed the suggestions of reviewer 1 and created new panels for Figure 1b (now Figure 1 b-d) and separated the panels by comparing PAC1R-PACAP27 versus VPAC1R-PACAP27 and VPAC1R-VIP versus VPAC1R-PACAP27. We did not include models of PAC1R-VIP in Figure 1, given that this is not an experimentally determined structure.

Given the high resolution of the structures obtained with the Gs-associated complexes, it might be worthwhile for the authors to offer some speculation on PAC1 versus VPAC1 coupling to Gq versus Gs, although this is perhaps a subject for another venue.

Author reply: We thank reviewer #1 for the suggestions, however we think that introducing and hypothesising on Gq and other G proteins is beyond the scope of the current manuscript, which is solely focused on Gs coupling, especially given the lack of an experimentally determined class B1 GPCR – Gq complexes to guide the predictions.

There are variant forms of the N-terminal ECD of PAC1. The authors should put their choice of PAC1n in context re: functional differences among variants.

Author reply: We have chosen the PAC1n variant, given that this variant is the most abundant and well-studied variant, and is also the most similar (in particular in PAC1R loop ICL3) to VPAC1R. The PAC1n variant is also selective for PACAP peptides over VIP peptides, whereas studies show that PAC1s has reduced selectivity (reference 16). We were aiming to resolve the PAC1Rn loop in the ECD, however, this loop is very flexible and only poorly resolved in our structures. We were able to include a backbone model of this loop for MD analysis to investigate possible interactions with other receptor or peptide residues. We have added the following at the start of the results section to explain the different splice variants and our choice of PAC1Rn splice isoform for our study.

“The PAC1R exists in different splice isoforms with a deletion within the ECD termed PAC1Rshort, and insertions within ICL3, termed PAC1Rhip, PAC1Rhop, PAC1Rhiphop [16]. PAC1Rnull does not have ICL3 insertions and contains additional residues in the ECD loop (residues 89-110) compared to PAC1Rshort [16]. For this study, PAC1R refers to the PAC1Rnull variant. This was selected as this is the most abundantly expressed variant physiologically, is the most well-studied, and the lack of ICL3 insertions provides the best comparison both structurally and pharmacologically to VPAC1R. This variant also has greater peptide selectivity of PACAP27 and PACAP38 over VIP relative to PAC1Rshort [16], and is therefore an ideal choice to provide insights into the selectivity of PAC1R for PACAP27 over VIP”.

Reviewer #2 (Remarks to the Author):

This is a solid paper, advancing our understanding on the important PACAP subfamily of proteins. Unfortunately, it is very hard to read and I recommend a major revision of the text.

Author reply: We thank reviewer #2 for the positive general comment on our paper. We made significant changes to the text in our manuscript to improve the readability, including moving some discussion points from the results to the discussion section and rewriting some sections to make the intended meaning clear. We have provided a tracked version of the revised manuscript, as well as a clean copy for review. We have also addressed the specific issues and examples pointed out by reviewer #2 as outlined below.

Examples:

“ Given the high degree of homology, there is significant interest in understanding the molecular basis for peptide selectivity between the 52 PAC1R and VPACRs”. If there would be a lower degree of homology, there would be no interest in understanding the molecular basis for peptide selectivity?”

Author reply: We have changed the sentence in the introduction to the following to make the intending meaning clear:

“These peptides have high homology with VIP, sharing nearly 70% sequence identity (**Fig. 1a**), and the PAC1R and VPAC1R also exhibit a high degree of sequence homology (56 % overall (<https://gpcrdb.org>)), which presents significant challenges for developing selective drugs. Thus, there is significant interest in understanding the molecular basis for peptide selectivity between the PAC1R and VPACRs.”

The cryoEM from the literature are mentioned as "static" structure and those solved by the authors not, although they are static just the same.

Author reply: We referred to other structures as static, given that all published structures of VPAC1R and PAC1R to date have not investigated any dynamic aspects, e.g. no 3D variability analysis was done. However, the reviewer is correct that structures themselves are static snapshots. We have conducted 3D variability analysis on all our datasets, giving us more information on the particle populations and particle dynamics, which we compare to our molecular dynamics data. We have revised all sections of the manuscript referring to static structures to refer to them as cryo-EM structures, rather than static structures, and we added clarifying sentences to distinguish the methods used in this paper that were not used in the published datasets, for example in the results section stating ‘Previously published complex structures of the VPAC family lack dynamic information that can be gained from cryo-EM datasets.’

“Binding to and activation of class B1 peptide GPCRs involves a complex series of sequential interactions that enables engagement with the peptide to overcome the energy barriers to activation”.

It is thermal fluctuations which allow to overcome free energy barriers, not sequential

interactions.

Author reply: This has been reworded to the following “Peptide binding to class B1 peptide GPCRs involves a complex series of sequential and dynamic interactions that enables initial peptide engagement with the receptor extracellular face to facilitate peptide N-terminal engagement deeper within the TM binding cavity where it can engage residues that are required to initiate conformational transitions required for receptor activation.”

“VIP makes fewer and less stable contacts with the receptor ECD in experimentally derived structure and/or models of interactions with active VPAC1R and PAC1R, and thus is more dynamic.” Why VIP should be necessarily more dynamic? VIP might be a much more rigid peptide than the others, even if it binds less strong to the receptor.

Author reply: We show in Figure 4 that the VIP peptide shows more variability in the molecular dynamics' frames than the PACAP27 peptide, therefore we interpret it as being more dynamic, in particular in the interactions of the C terminus. The intended meaning of this sentence was that receptor interactions are more dynamic, but we were not implying that it is a more or less rigid peptide helix. Nonetheless, we have removed the “and is thus more dynamic” from the sentence so the meaning is not confused.

Reviewer #3 (Remarks to the Author):

In this paper, Wootten et al present three cryo-EM structures of VPAC receptors, VPAC1R-VIP, VPAC1R-PACAP27, and PAC1R-PACAP27 complexes. Through structural analysis, they proposed peptide ligand binding specificity. In combination with MD modeling, they showed that VIP has fewer interaction with PAC1R and VPAC1R than for PACAP27 and is more dynamic in the PAC1R-VIP complex than for the PAC1R-PACAP27 complex. The structural works are built on the previous structures of VPAC1R-VIP and PAC1R-PACAP38 complexes. The structures presented here are of better resolutions and are with water molecules, provide additional insights into peptide ligand recognition.

Author reply: We thank reviewer #3 for the positive feedback on our manuscript. However, we would like to clarify that there is no previous study that presents a structure of VPAC1R-VIP - our manuscript presents the first structure of VPAC1R-VIP. We assume that reviewer #3 is referring to the VPAC1R-PACAP27 structure published by Duan et al in 2020 (also addressed below).

The most important point is the selectivity of VIP for VPAC1R over PAC1R. However, the specific residues for such selectivity between these two receptors are not clearly presented. Additional mutations that can switch specificity between VPAC1R and PAC1R should be demonstrated to fully nail down the residues in the receptors for the ligand specificity

There are only nine different residues between VIP and PACAP27. The relative contribution of each residue to the specificity for VPAC1R should be demonstrated by swapping each different residue between VIP and PACAP27.

Author reply: We acknowledge the point that reviewer #3 makes in regards to peptide selectivity and creating chimera peptides as well as chimera receptors. There is already some literature that investigates these points, and we have further investigated the existing literature and added in additional text throughout the results and discussion (with relevant references) to highlight that published work, which addresses some of these points. Testing mutants/chimera peptides in which all non-conserved residues are swapped between PACAP27 and VIP is beyond the scope of this manuscript. However, as we hypothesised from our experiments that the residue at position 4 is one of the key drivers of peptide selectivity, we did generate two new peptides - chimeras swapping Gly/Ala4 between the PACAP27 and VIP. We performed cAMP assays for both receptors which revealed that these swaps had no significant impact on VPAC1R-mediated cAMP production, however introduction of Ala4 into PACAP27 significantly reduced the potency of PACAP27 for the PAC1R, whereas introduction of Gly at position 4 significantly enhanced the potency of VIP for the PAC1R. We also assessed these peptides in PAC1R whole cell binding studies with similar trends (albeit the Ala4-PACAP27 effect was much larger than the Gly4-VIP). These data have been added into the manuscript, with text added to the results and discussion.

It is beyond the scope of this manuscript to change all potential residues where we observed differences in receptor engagement residues between the VPAC1 and PAC1 receptors. In addition to non-conserved residues, we identified that many conserved residues (both on the peptide and receptor, and in pair-wise interactions) engage differently between the two receptors (e.g. in occupancy frames in the MD simulation, or in the unbinding/binding experiments); such differences are only revealed in analysis of the complex dynamics.

Because of this, we believe that single site mutational analysis would likely be difficult to interpret in this context. Nonetheless, as noted above, we have added in reference to published mutagenesis data, where it is available, which supports our structural and computational studies.

PACAP27 and PACAP38 can bind both VPAC1R and PAC1R well. The peptide binding energy of these peptides to these receptors should be analyzed to support their binding properties.

Author reply: We thank the reviewer for the suggestion. We have investigated peptide binding energies and added these into our supplementary data and table and refer to these in the manuscript results section.

Results section: “Binding energies of the peptides to the receptors are summarized in **Supplementary Table 9** and show that the PACAP27 peptide, when bound to the VPAC1R or PAC1R, has similar binding energies, whereas the VIP-PAC1R complex has less negative energy, which is in line with the structural and pharmacology data in this paper, showing that the lower affinity peptide VIP forms a less stable complex with PAC1R. Per-residue contributions to the binding energies are also summarised in **Supplementary Fig. 12**, in which charged residues of the mid-region of the peptide (R12, R14) show strong negative binding energies, particularly for the PAC1R bound peptides.”

Results section: “Calculation of peptide binding energies (**Supplementary Table 9**) also show more negative energy for the VIP-PAC1R complex in the presence of the disulphide bond, suggesting that the ECD-ECL1 disulphide stabilises the VIP-bound complex.”

The structures of VPAC1R-VIP and PAC1R-PACAP28 complexes have been reported previously. Detailed structure comparison between the previous structures and the current structures should be presented with RMSD between these structures for the receptor and for the peptide ligands.

Author reply: We thank the reviewer for the suggestion. We have added a supplementary figure and table to compare all previously published structures with the structures in this manuscript. Reviewer #3 refers to VPAC1R-VIP, however, this structure is reported in this manuscript for the first time. We assume that the reviewer is actually referring to the published VPAC1R-PACAP27 structure by Duan et al. The following has been added to the results section: “Recent cryo-EM structures of PAC1R-PACAP38 (PDB: 6M1I [11], PDB: 6LPB [22], PDB: 6P9Y [9]) at 3.5 Å, 3.9 Å and 3.01 Å, respectively, are in overall agreement with the presented PAC1R-PACAP27 in terms of secondary structure comparison of the models built into the cryo-EM maps (**Supplementary Table 1**). However, previous cryo-EM maps lack the resolution to model water molecules, and extracellular domains and loops could also not be modelled (**Supplementary Fig. 6a**)”.

The previous structure of VPAC1R-VIP complex was determined with Nanobit tethering method for stabilizing the G protein complex, where the current VPAC1R-VIP structure is not. The bound Gs heterotrimer between these two structures should be analyzed to see whether there are any differences.

Author reply: In conjunction with the previous comment, we have added a supplementary figure to compare all previously published structures with the structures in this manuscript. As described above, we assume that the reviewer is actually referring to the published VPAC1R-PACAP27 structure by Duan et al. In the structure comparison figure, we compared the Gs protein position and RMSD in our structures without Nanobit tethering to the published VPAC1R-PACAP27 structure with nanobit tethering. As shown in Movie 1 and Figure 1B, there are slight offsets of G protein positions (when structures are receptor-aligned), which are sampled in the conserved rocking and twisting motions of the receptor- G protein complex dynamics for all Class B1 receptor complexes. Nevertheless, the VPAC1R-PACAP27 structures with and without nanobit tethering are in high overall agreement, except for ECL1 and the ECD that were not modelled in the previous VPAC1R-PACAP27 nanobit structure. We have added a comment about the previous VPAC1R-PACAP27 structure using the Nanobit technology, and a figure and table comparing the structures with RMSD for Gs, receptor and peptide chains as outlined below

Results section: “The published structure of VPAC1R-PACAP27 at 3.2 Å (PDB 6VN7) used the Nanobit technology to stabilise the G protein complex [10], and the model for this also aligns well at the secondary structure level with our VPAC1R-PACAP27 structure (**Supplementary Fig. 6b**).”

REVIEWERS' COMMENTS

Reviewer #3 (Remarks to the Author):

The authors have done a great job in addressing my comments and I fully support the publication of this interesting paper.

REPONSE TO REVIEWER COMMENTS

We thank the reviewers for the review of our work. No further changes to the manuscript were requested from the reviewers.

structures are receptor-aligned), which are sampled in the conserved rocking and twisting motions of the receptor- G protein complex dynamics for all Class B1 receptor complexes. Nevertheless, the VPAC1R-PACAP27 structures with and without nanobit tethering are in high overall agreement, except for ECL1 and the ECD that were not modelled in the previous VPAC1R-PACAP27 nanobit structure. We have added a comment about the previous VPAC1R-PACAP27 structure using the Nanobit technology, and a figure and table comparing the structures with RMSD for Gs, receptor and peptide chains as outlined below

Results section: “The published structure of VPAC1R-PACAP27 at 3.2 Å (PDB 6VN7) used the Nanobit technology to stabilise the G protein complex [10], and the model for this also aligns well at the secondary structure level with our VPAC1R-PACAP27 structure (**Supplementary Fig. 6b**).”